# Towards IID representation learning and its application on biomedical data

**Jiqing Wu**[1]                         JIQING.WU@USZ.CH
**Inti Zlobec**[2]                  INTI.ZLOBEC@PATHOLOGY.UNIBE.CH
**Maxime Lafarge**[1]                    MAXIME.LAFARGE@USZ.CH
**Yukun He**[3]                       YUKUNHE@CITYU.EDU.HK
**Viktor H. Koelzer**[1,4]                  VIKTOR.KOELZER@USZ.CH

[1] *Department of Pathology and Molecular Pathology, University Hospital, University of Zurich, Switzerland.*

[2] *Department of Pathology, University of Bern, Switzerland.*

[3] *Department of Mathematics, City University of Hong Kong, China.*

[4] *Department of Oncology and Nuffield Department of Medicine, University of Oxford, UK.*

**Editors:** Under Review for MIDL 2022

## Abstract

Due to the heterogeneity of real-world data, the widely accepted *independent and identically distributed* (IID) assumption has been criticized in recent studies on causality. In this paper, we argue that instead of being a questionable assumption, IID is a fundamental task-relevant property that needs to be learned. Consider $k$ independent random vectors $\mathsf{X}^{i=1,\dots,k}$, we elaborate on how a variety of different causal questions can be reformulated to learning a task-relevant function $\phi$ that induces IID among $\mathsf{Z}^i := \phi \circ \mathsf{X}^i$, which we term IID representation learning.

For proof of concept, we examine the IID representation learning on Out-of-Distribution (OOD) generalization tasks. Concretely, by utilizing the representation obtained via the learned function that induces IID, we conduct prediction of molecular characteristics (molecular prediction) on two biomedical datasets with real-world distribution shifts introduced by a) preanalytical variation and b) sampling protocol. To enable reproducibility and for comparison to the state-of-the-art (SOTA) methods, this is done by following the OOD benchmarking guidelines recommended from WILDS. Compared to the SOTA baselines supported in WILDS, the results confirm the superior performance of IID representation learning on OOD tasks. The code is publicly accessible via https://github.com/CTPLab/IID_representation_learning.

**Keywords:** IID, IID representation learning, OOD generalization, causality, biomedical.

## 1. Introduction

In machine learning (Vapnik, 1999), we commonly assume that data entries $(y_i, \boldsymbol{x}_i)_{i=1,\dots,n}$ are independently drawn from the same probability distribution $\mathbb{P}_{(\mathsf{Y},\mathsf{X})}$ of a random vector $(\mathsf{Y}, \mathsf{X})$. This is referred to as the *independent and identically distributed* (IID) assumption. However, real-world data is usually characterized by significant heterogeneity (Bareinboim, 2014; Peters et al., 2017; Arjovsky et al., 2019; Rosenfeld et al., 2021). Controlling data heterogeneity is particularly critical in application of data driven methods to the medical domain (Cios and Moore, 2002), as medical algorithms that suffer from prediction degradation on heterogeneous cohorts can have severe consequences in medical practice. Consequently, the IID assumption needs to be critically questioned.

The task of learning a robust model that is resistant to a heterogeneous data distribution is formally denoted as Out-of-Distribution generalization (OOD) (Arjovsky et al., 2019; Koh et al., 2021). For a thorough overview we refer interested readers to (Shen et al., 2021). A large number of studies with diverse methodologies  (Peters et al., 2016; Ganin et al., 2016; Sun and Saenko, 2016; Rojas-Carulla et al., 2018; Arjovsky et al., 2019; Sagawa et al., 2020; Rosenfeld et al., 2021) have been proposed to address this issue. From the viewpoint of domain adaptation (Pan et al., 2010), the root causes of OOD failure come from domain or task shift (Wang and Deng, 2018). There have been many studies dedicated to resolve the challenge. (Sun et al., 2016; Sun and Saenko, 2016) proposed to align the second-order statistics of the source and target distributions. In case of simultaneous domain and task shift, (Gong et al., 2016) suggested to pinpoint conditional transferable components. Further, (Long et al., 2018) reduced the shifts in the data distributions across domains via adversarial learning (Goodfellow et al., 2014). Built upon the invariant property reflected in causality (Pearl et al., 2000), (Peters et al., 2016) firstly proposed the seminal invariant causal prediction (ICP) framework. Later, (Rojas-Carulla et al., 2018) investigated the invariant set and extended the ICP to transfer learning (Pan and Yang, 2009; Muandet et al., 2013; Zhuang et al., 2020). Motivated by the ICP, invariant risk minimization (IRM) (Arjovsky et al., 2019) was subsequently proposed to learn an invariant predictor that is optimal for all environments. Recently, (Schölkopf et al., 2021) pointed out the essential role of causal representation learning in OOD generalization. In a nutshell, (Schölkopf et al., 2021) argued that cause-effect relations are critical components of reasoning chains that remain robust in situations beyond training tasks. However, causal variables are usually not given in machine learning tasks. Thus, (Schölkopf et al., 2021) suggested learning causal representations to resolve the limitation of current approaches for OOD generalization.

Inspired by impactful studies centered on the investigation of statistical invariance:

- We introduce a novel pair of definitions: IID symmetry and its generalization. These definitions reflect the core message delivered in the work, *i.e.*, instead of being a questionable assumption, IID is a fundamental task-relevant property that needs to be learned.

- Then, we systematically discuss how IID and causality are two sides to the same coin. Consider $k$ independent random vectors $\mathsf{X}^{i=1,\dots,k}$, we elaborate concrete examples of reformulating diverse causal problems to learning a task-relevant function $\phi$ that induces IID among $\mathsf{Z}^i := \phi \circ \mathsf{X}^i$, which we term IID representation learning.

- For proof of concept, we examine the IID representation learning on Out-of-Distribution (OOD) generalization tasks. Concretely, in utilizing the representation obtained via the learned function that induces IID, we conduct molecular prediction experiments on two comprehensive biomedical datasets (RxRx1 (Taylor et al., 2019) and Swiss Colorectal Cancer (SCRC) (Nguyen et al., 2021)). By following the OOD benchmarking guidelines recommended from WILDS (Koh et al., 2021), we demonstrate that the IID representation learning can improve the molecular predictions compared to the SOTA baselines supported in WILDS.

## 2. Proposed Definition

As elaborated above, the common ground of causal studies usually starts with exploring statistical invariance. Thus, we introduce the definitions of IID symmetry and its generalization as follows: Consider $k + n$ independent random vectors $\mathsf{X}^1, \dots, \mathsf{X}^k, \mathsf{X}^{k+1}, \dots, \mathsf{X}^{k+n}$ and a Lebesgue integrable

$\phi : \mathbb{R}^{l+1} \mapsto \mathbb{R}^{m+1}$, for $i = 1, \ldots, k + n$, let $\mathbb{Q}_{X^i}$ be a query distribution[1] of $X^i = (x_0^i, x_1^i, \ldots, x_l^i)$, let $Z^i = (z_0^i, z_1^i, \ldots, z_m^i) := \phi \circ X^i$ and $\mathbb{Q}_{Z^i} := \mathbb{Q}_{X^i} \circ \phi^{-1}$,

**Definition 1** *We say that* $X^1, \ldots, X^k$ *have an ($\phi-$)**IID symmetry** if $\phi$ induced $\mathbb{Q}_{Z^1}, \ldots, \mathbb{Q}_{Z^k}$ are identical distributions, i.e., $\mathbb{Q}_{Z^1} = \ldots = \mathbb{Q}_{Z^k}$. Further, we say that the ($\phi-$)IID symmetry is **generalizable** to* $X^{k+1}, \ldots, X^{k+n}$ *if $\mathbb{Q}_{Z^1}, \ldots, \mathbb{Q}_{Z^k}, \mathbb{Q}_{Z^{k+1}}, \ldots, \mathbb{Q}_{Z^{k+n}}$ are identical distributions.*

**Remark 1** *It is not difficult to see that* $Z^1, \ldots, Z^{k+n}$ *are independent, since w.l.o.g. we can reduce the proof to the simpler case of two random vectors* $Z^1, Z^2$ *and $\phi$ being continuous. Let $f : \mathbb{R}^{m+1} \to \mathbb{R}$ be bounded and continuous, then $f \circ \phi : \mathbb{R}^{l+1} \to \mathbb{R}$ is also bounded and continuous. We have*

$$
\begin{aligned}
\mathbb{E}[f(Z^1)f(Z^2)] &= \mathbb{E}[(f \circ \phi)(X^1)(f \circ \phi)(X^2)] = \mathbb{E}[f \circ \phi(X^1)]\mathbb{E}[f \circ \phi(X^2)] \\
&= \mathbb{E}[f(Z^1)]\mathbb{E}[fZ^2)],
\end{aligned}
\tag{1}
$$

*where the second equality comes from the independence of* $X^1$ *and* $X^2$. *As a large class of functions including piece-wise continuous function (neural network) satisfies the Lebesgue integrability condition, we claim the map $\phi$ discussed in this paper always induces independence. Since for $i = 1, \ldots, k + n$, $Z^i$ is independent and identically distributed w.r.t.* $\mathbb{Q}_{\phi \circ X^i}$, *we call $Z^i$ an ($\phi-$)**IID representation**. It is worth mentioning that the entries of $Z^i$ are not required to be independent.*

**Remark 2** *For $i = 1, \ldots, k + n$, if $\mathbb{Q}_{X^i} = \mathbb{P}_{X^i}$ is the probability distribution of $X^i$, then $Z^1, \ldots, Z^{k+n}$ are IID in the canonical sense according to Rem. 1. Besides, the **trivial** IID symmetry and its generalization always exist, for instance we can define a **trivial** $\phi$ such that $\phi \circ X^i = const$.*

## 3. From Causality to IID

Causal inference is a fundamental research domain that reflects the zeitgeist in machine learning (Luo et al., 2020). Broadly speaking, prior studies on causal inference can be categorized into two areas of research: causal identification (Pearl et al., 2009; Peters et al., 2017; Hernán and Robins, 2020) and causal transportation (Balke and Pearl, 1995; Bareinboim and Pearl, 2014; Bareinboim, 2014). The former aims to either identify the underlying Structural Causal Models (SCM) (Peters et al., 2017) or quantify the Average Causal Effect (ACE) (Hernán and Robins, 2020), whereas the latter is often meant for licensing the transportable causal knowledge from one population to another (Bareinboim and Pearl, 2014; Bareinboim, 2014). In a recent study (Schölkopf et al., 2021), the authors propose causal representation learning to resolve OOD generalization. To link causal inference and IID, we first introduce two prerequisite concepts:

**Structural Causal Model.** Following the specification in (Peters et al., 2016, 2017), consider a Structural Causal model (SCM), *i.e.*, there exists a random vector $X = (x_0, \ldots, x_l)$ and a directed acyclic graphs (DAG) consisting of vertices $x_0, \ldots, x_l$ and $\delta_0, \ldots, \delta_l$ such that for $j = 0, \ldots, l$ we have

$$
x_j = f_j(X_{PA_j}, \delta_j), \delta_j \perp\!\!\!\perp X_{PA_j},
\tag{2}
$$

where $X_{PA_j} \subset \{x_0, x_1, \ldots, x_l\}$ is the set of known parents of $x_j$, $\delta_j$ is the unknown (parent) noise. By drawing arrow(s) from $X_{PA_j}, \delta_j$ to $x_j$ defined in Eq. 2, we obtain the edges of the DAG (See Fig. 1

---

1. *i.e.*, a probability measure induced by $X^i$ (*e.g.*, conditional, marginal, and probability distribution of $X^i$) that is relevant to the question of interest.

for graph visualization). The SCM bears many practical interests for analyzing complex medical datasets, *e.g.*, given the patient overall survival $x_0 := x_{os}$, we want to identify the key prognostic variables among $x_{age}, x_{gender}, x_{BMI}$, *etc.* that directly impact $x_{os}$ (Shapiro and Msaouel, 2021).

**Do-Intervention**. As discussed in (Pearl and Mackenzie, 2018), one of the most prominent building blocks of causal inference is intervention. Formally, we denote the (hard) do-intervention, *i.e.*, the replacement of Eq. 2 with $x_j :=$ const by $do(x_j =$ const$)$. Noting that intervening on $x_j$ breaks the arrow(s) between $X_{PA_j}, \delta_j$ and $x_j$. Accordingly, we denote the interventional distribution of $x_0$ conditioned on $x_1, \ldots, do(x_j =$ const$), \ldots, x_l$ by $\mathbb{P}(x_0 \mid x_1, \ldots, x_j^e, \ldots, x_l)$, the random vector by $X^e = (x_0, x_1, \ldots, x_j^e, \ldots, x_l)$ and the set of known parents of $x_j$ by $X_{PA_j}^e$. In the clinical domain, it should be noted that the implementation of do-intervention is expensive (Martin et al., 2017) owing to regulatory scrutiny and ethically challenging. This is illustrated by recent publications critically discussing such interventions as placebo surgery (Angelos, 2013) and the involvement of vulnerable patient groups (Caldwell et al., 2004; Farrell et al., 2020), *etc.*

In real-world applications, randomized clinical trials (RCT) are considered to be the gold-standard for interventional clinical studies (Nout et al., 2010; de Boer et al., 2019). Given the patient outcome $x_0$, we are keen on understanding the distribution of $x_0$ conditioned on (intervened) treatment $x_1$ and prognostic variables $x_2, \ldots, x_l$ in the presence of unknown noises. Thus, we discuss how various related causal problems can be reformulated to learning a function inducing IID.

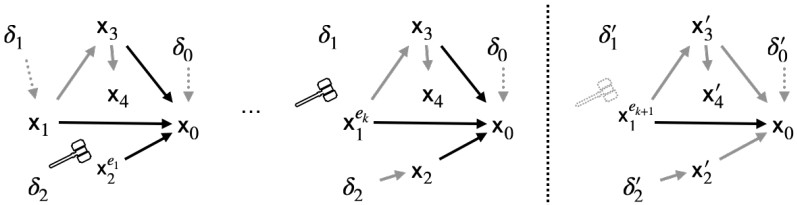

Figure 1: Left: The graphical visualization for causal variable identification. The black arrows indicate identical distributions $\mathbb{P}(x_0 \mid X_{PA_0}^{e_1}) = \ldots = \mathbb{P}(x_0 \mid X_{PA_0}^{e_k})$, the dotted arrows connect the unknown noises. The black hammers indicate the do-interventions $e_1, \ldots, e_k$ implemented in the form of RCTs. Right: The graphical visualization for causal effect transportation. The black arrow indicates that the distribution $\mathbb{P}'(x_0 \mid x_1^{e_{k+1}})$ is transported from the identical $\mathbb{P}(x_0 \mid x_1^{e_k})$, where the gray dotted hammer indicates the do-intervention $e_{k+1}$ that leads to $\mathbb{P}'(x_0 \mid x_1^{e_{k+1}})$ and cannot be implemented in the setting of a RCT due to ethical reasons.

### 3.1. Causal Variable Identification $\rightarrow$ IID symmetry

Let us assume $k$ SCMs underlying a medical datatset collected from clinical trials, *i.e.*, for $i = 1, \ldots, k$ there exists an $X^{e_i} = (x_0, x_1, \ldots, x_{j_1}^{e_i}, \ldots, x_{j_{e_i}}^{e_i}, \ldots, x_l)$ and its corresponding DAG with unknown noises $\delta_0, \ldots, \delta_l$, where $x_0$ is the patient outcome, $e$ represents the do-intervention(s) imposed on a subset variables of $\{x_1, \ldots, x_l\}$ in $X$ (See Fig. 1 (left)). Due to the NP-hard challenge of learning an entire DAG (Chickering, 1996; Luo et al., 2020), invariant causal prediction (ICP) (Peters et al., 2016) was proposed to identify plausible causal variables given the outcome of interest (here patient outcome $x_0$). Since for $i = 1, \ldots, k$, $X_{PA_0}^{e_i}$ is the set of plausible causal variables of $x_0$ (Peters et al., 2016), under the assumption of identical interventional distributions $\mathbb{P}(x_0 \mid X_{PA_0}^{e_i})$ brought by $k$ different do-interventions we propose:

**Question** *Consider $k$ independent random vectors $\mathsf{X}^{e_1}, \ldots, \mathsf{X}^{e_k}$ specified above, for $i = 1, \ldots, k$ let $\mathbb{Q}_{\mathsf{X}^{e_i}} = \mathbb{P}(\mathsf{x}_0 \mid \mathsf{x}_1, \ldots, \mathsf{x}^{e_i}_{j_1}, \ldots, \mathsf{x}^{e_i}_{j_{e_i}}, \ldots, \mathsf{x}_l)$, can we find a $\phi$ in Def. 1 such that $\mathbb{Q}_{\phi \circ \mathsf{X}^{e_1}}, \ldots, \mathbb{Q}_{\phi \circ \mathsf{X}^{e_k}}$ are identical distributions and it satisfies $\phi(x_0, x_1, \ldots, x_l) = (x_0, \ldots)$?*

**Discussion** *The map $\phi \circ \mathsf{X}^{e_i} = (\mathsf{x}_0, \mathsf{X}^{e_i}_{\mathsf{PA}_0})$ that projects $(\mathsf{x}_0, \mathsf{x}_1, \ldots, \mathsf{x}^{e_i}_{j_1}, \ldots, \mathsf{x}^{e_i}_{j_{e_i}}, \ldots, \mathsf{x}_l)$ to $(\mathsf{x}_0, \mathsf{X}^{e_i}_{\mathsf{PA}_0})$ induces the identical $\mathbb{Q}_{\phi \circ \mathsf{X}^{e_i}} = \mathbb{P}(\mathsf{x}_0 \mid \mathsf{X}^{e_i}_{\mathsf{PA}_0})$. This is the consequence of Eq. 2, since for $i = 1, \ldots, k$ the assignment $f_0$ between $\mathsf{x}_0$ and $\mathsf{X}^{e_i}_{\mathsf{PA}_0}, \delta_0$ remains unchanged and $\delta_0$ is independent of $\mathsf{X}^{e_i}_{\mathsf{PA}_0}$. In the toy experiments (App. A), we demonstrate the robustness of learning a projection map inducing identical interventional distributions, where the map is parametrized with a simple neural network.*

### 3.2. Causal Effect Transportation → IID Generalization

Consider for $i = 1, \ldots, k$, we know the assignment $f_0$ between $\mathsf{x}_0$ and $\mathsf{X}^{e_i}_{\mathsf{PA}_0}$ (Eq. 2) *w.r.t.* the identical $\mathbb{P}(\mathsf{x}_0 \mid \mathsf{X}^{e_i}_{\mathsf{PA}_0})$, since it is unethical and infeasible to re-run the clinical trial on lots of patient cohorts, we often want to transport the causal knowledge to a new observational cohort (Bareinboim, 2014). Let $\mathsf{X}^{k+1} = (\mathsf{x}_0, \mathsf{x}_1, \mathsf{x}'_2, \ldots, \mathsf{x}'_l)$ be a random vector representing the observational cohort, based on the causal knowledge learned by $\mathsf{X}^{e_1}, \ldots, \mathsf{X}^{e_k}$, we aim to compute $\mathbb{P}'(\mathsf{x}_0 \mid \mathsf{x}^{e_{k+1}}_1)$ of $\mathsf{X}^{e_{k+1}} = (\mathsf{x}_0, \mathsf{x}^{e_{k+1}}_1, \mathsf{x}'_2, \ldots, \mathsf{x}'_l)$ (Bareinboim, 2014), *i.e.*, the distribution of patient outcome $\mathsf{x}_0$ conditioned on the intervened treatment $\mathsf{x}^{e_{k+1}}_1$, Under the assumption of identical interventional distributions brought by $k + 1$ different do-interventions we propose:

**Question** *Consider $k$ independent random vectors $\mathsf{X}^{e_1}, \ldots, \mathsf{X}^{e_k}$ specified in Sec. 3.1, for $i = 1, \ldots, k$ let $\mathbb{Q}_{\mathsf{X}^{e_i}} = \mathbb{P}(\mathsf{x}_0 \mid \mathsf{X}^{e_i}_{\mathsf{PA}_0})$, we further assume an $\mathsf{X}^{e_{k+1}} = (\mathsf{x}_0, \mathsf{x}^{e_{k+1}}_1, \mathsf{x}'_2, \ldots, \mathsf{x}'_l)$ independent of $\mathsf{X}^{e_1}, \ldots, \mathsf{X}^{e_k}$ and $\mathbb{Q}_{\mathsf{X}^{e_{k+1}}} = \mathbb{P}'(\mathsf{x}_0 \mid \mathsf{X}^{e_{k+1}}_{\mathsf{PA}_0})$, can we find a $\phi$ in Def. 1 such that $\mathbb{Q}_{\phi \circ \mathsf{X}^{e_1}}, \ldots, \mathbb{Q}_{\phi \circ \mathsf{X}^{e_k}}, \mathbb{Q}_{\phi \circ \mathsf{X}^{e_{k+1}}}$ are identical distributions and it satisfies $\phi(x_0, x_{\mathsf{PA}_0}) = (x_0, x_1, \ldots)$?*

**Discussion** *If the patient outcome conditioned on the intervened treatment remains invariant across different cohorts, by determining $\phi \circ \mathsf{X}^{e_{k+1}}_{\mathsf{PA}_0} = (\mathsf{x}_0, \mathsf{x}^{e_{k+1}}_1)$ we have $\mathbb{Q}_{\phi \circ \mathsf{X}^{e_1}} = \ldots = \mathbb{Q}_{\phi \circ \mathsf{X}^{e_{k+1}}} = \mathbb{P}'(\mathsf{x}_0 \mid \mathsf{x}^{e_{k+1}}_1)$ (See Fig. 1 (right)). Otherwise if the patient outcome conditioned on the intervened treatment in the same age group ($\mathsf{x}'_2 := \mathsf{x}_{\mathsf{age}}$) remains invariant, then we need to derive $\phi \circ \mathsf{X}^{e_{k+1}}_{\mathsf{PA}_0} = (\mathsf{x}_0, \mathsf{x}^{e_{k+1}}_1, \mathsf{x}'_2)$ and obtain $\mathbb{Q}_{\phi \circ \mathsf{X}^{e_1}} = \ldots = \mathbb{Q}_{\phi \circ \mathsf{X}^{e_{k+1}}} = \mathbb{P}'(\mathsf{x}_0 \mid \mathsf{x}^{e_{k+1}}_1, \mathsf{x}_{\mathsf{age}})$, thus we conclude $\mathbb{P}'(\mathsf{x}_0 \mid \mathsf{x}^{e_{k+1}}_1) = \sum \mathbb{P}'(\mathsf{x}_0 \mid \mathsf{x}^{e_{k+1}}_1, \mathsf{x}_{\mathsf{age}}) \mathbb{P}'(\mathsf{x}_{\mathsf{age}})$, where $\mathbb{P}'(\mathsf{x}_{\mathsf{age}})$ is the marginal distribution of $\mathsf{x}_{\mathsf{age}}$.*

### 3.3. Causal Feature Representation → IID Representation

One of the open questions raised in (Schölkopf et al., 2021) is how to learn a reusable feature representation of $\mathsf{X} = (\mathsf{x}_1, \ldots, \mathsf{x}_l)$. This question becomes essential when $\mathsf{x}_1 \ldots, \mathsf{x}_l$ do not correspond to well-studied treatment and prognostic variables, but to pixels of medical imaging data that bear critical information of possibly unknown variables. Based on the Independent Causal Mechanism (ICM) (Peters et al., 2017) and Sparse Mechanism Shift (SMS), (Schölkopf et al., 2021) hypothesize that learning a causal-aware representation in an auto-encoder fashion is promising for its reusability in downstream tasks. In alignment with this keen insight and the assumption that latent representations of training, validation and test datasets have identical probability distributions:

**Question** *Consider $k+n+p$ independent random vectors $\mathsf{X}^1, \ldots, \mathsf{X}^k, \mathsf{X}^{k+1}, \ldots, \mathsf{X}^{k+n}, \mathsf{X}^{k+n+1}, \ldots, \mathsf{X}^{k+n+p}$, for $i = 1, \ldots, k + n + p$ let $\mathbb{Q}_{\mathsf{X}^i} = \mathbb{P}_{\mathsf{X}^i}$ be the probability distribution of $\mathsf{X}^i$, can we find a $\phi$ in Def. 1*

such that $\mathbb{Q}_{\phi \circ X^1}, \ldots \mathbb{Q}_{\phi \circ X^{k+n+p}}$ are identical distributions and there exists a $\phi' : \mathbb{R}^m \mapsto \mathbb{R}^l$ satisfying $\phi' \circ \phi = \mathsf{id}$?

**Discussion** *According to Rem. 1, 2, we aim to learn an IID representation $Z^i = \phi \circ X^i = (z_1^i, \ldots, z_m^i)$ for $i = 1, \ldots, k + n + p$ as if the images in training ($X^1, \ldots, X^k$), validation ($X^{k+1}, \ldots, X^{k+n}$) and test ($X^{k+n+1}, \ldots, X^{k+n+p}$) datasets can be faithfully reconstructed from the identical distribution $\mathbb{P}_{Z^i}$. In the following experiments, we demonstrate the reusability of learned IID representation for downstream prediction tasks.*

## 4. OOD Experiment

As discussed above, one of the biggest challenges in application of machine learning methodologies to the medical domain lies in data heterogeneity that violates the conventional IID assumption. There are many factors contributing to the heterogeneity such as preanalytical variation (Taylor et al., 2019), sampling protocol (Karamitopoulou et al., 2011), *etc*. As the goal of OOD generalization is to resolve the challenge of heterogeneous training and test data (Shen et al., 2021), we examine the IID representation learning under the OOD setting and conduct prediction of molecular characteristics (molecular prediction) on two comprehensive biomedical datasets–RxRx1 (Taylor et al., 2019) and Swiss Colorectal Cancer (SCRC) (Nguyen et al., 2021). The former aims to predict genetic perturbations given fluorescence microscopy images of cancer cells contaminated with preanalytical batch effects, while the latter study aims to classify the consensus molecular subtypes (imCMS1-4 (Sirinukunwattana et al., 2020)) of colorectal cancer (CRC) based on tissue microarray (TMA) images, where the TMAs are heterogeneously sampled from different tumor regions.

To enable reproducibility and for comparison to the SOTA methods, we run molecular prediction experiments by following the guidelines of WILDS (Koh et al., 2021). Accordingly, we split RxRx1 to training (40612 images), validation (9854), in-distribution (ID) (40612) and OOD test (34432) data. Since SCRC contains TMAs sampled from tumor front (3333), micro-environment (micro) (2819) and center (3914) regions, we take images from two out of the three tumor regions to form the training data. By excluding 2 TMAs/patient from the held-back region as validation, we have the remaining TMAs as OOD test data. This leads to three variants of experiments: SCRC0 (front and micro for training), 1 (micro and center for training) and 2 (center and front for training). We then compare the IID representation learning to the SOTA baselines supported in WILDS: Empirical risk minimization (ERM) that minimizes the average classification loss on training sample (Vapnik, 1992; Shen et al., 2021), invariant risk minimization (IRM) (Arjovsky et al., 2019) with ERM + gradient regularization, correlation alignment (CORAL) (Sun and Saenko, 2016) with ERM + covariance regularization, group distributed robust optimization (GroupDRO) (Sagawa et al., 2020) with ERM + worst-case group regularization. For the IID representation learning, we first learn an IID representation in an auto-encoder fashion and then combine the learned IID representation with ERM for downstream molecular predictions, *i.e*., ERM + IID representation (See Fig. 2). According to WILDS' experiment and metric design, all molecular prediction experiments are run at least 3 times (4 times in our case) and we report average prediction results with standard deviation (SD).

**Learning the Approximate IID Representation**. Despite being conceptually simple, learning an IID representation that can faithfully reconstruct a given input image is non-trivial. To approximate the IID property and to achieve good reconstruction quality, we propose to utilize the instance normalization (IN) (Ulyanov et al., 2016) in the encoder for proof of concept. Concretely, we apply two kinds of blocks containing IN operations: morphology (morph) and stain to obtain

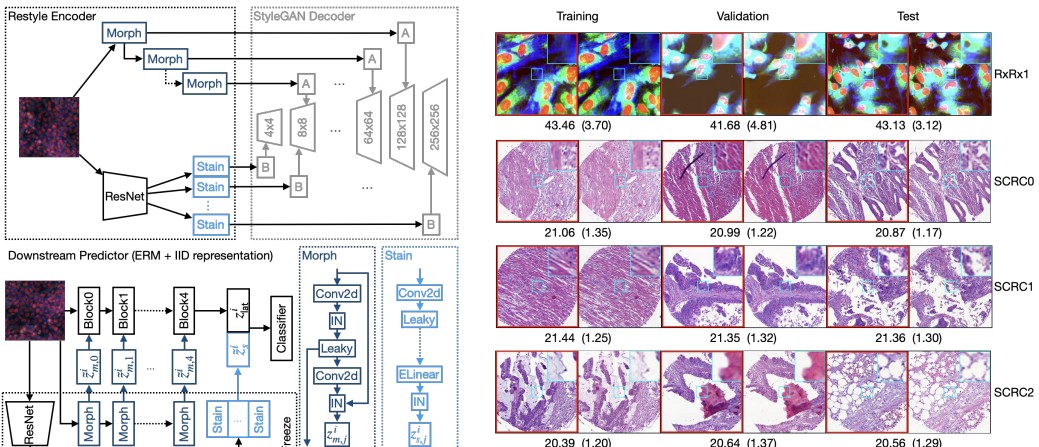

Figure 2: Left: The model illustrations of the proposed IID representation learning (Restyle Encoder and StyleGAN Decoder (Alaluf et al., 2021)) and its downstream molecular predictor (ERM + IID representation). Right: The visual comparison and average PSNR with SD achieved by the IID representation learning. Here, we normalize the RxRx1 images along each channel and zoom in on a small region of ground-truth (red bounding box) and reconstructed images for better visualization.

$$\mathsf{Z}^i := (\overbrace{\mathsf{Z}^i_{m,0}, \ldots, \mathsf{Z}^i_{m,4}}^{\text{morph}}, \overbrace{\mathsf{Z}^i_{s,0}, \ldots, \mathsf{Z}^i_{s,13}}^{\text{stain}})$$ in Sec. 3.3 (See Fig. 2). Compared to other normalization strategies (Ioffe and Szegedy, 2015; Ba et al., 2016), IN allows to impose the identical mean and standard deviation on the entries of $\mathsf{Z}^i_{m,0}, \ldots, \mathsf{Z}^i_{m,4}, \mathsf{Z}^i_{s,0}, \ldots, \mathsf{Z}^i_{s,13}$ without violating the independence of $\mathsf{Z}^i$ (See App. C for more normalization studies). This suggests that the learned representation $\mathsf{Z}^i$ is independent and approximately identically distributed.

Based on the recent development in image inversion (Alaluf et al., 2021), we instantiate the $\phi, \phi'$ in Sec. 3.3 with the Restyle encoder (Alaluf et al., 2021) and StyleGAN decoder (Karras et al., 2020). As shown in Fig. 2 (left), we couple the morph and stain with the noise (A) and style (B) modules of StyleGAN respectively. This is meant for learning a semantic-aware representation for the follow-up interpretation (See Fig. 11). Then, the objective is to reconstruct the input image with $256 \times 256$ resolution and defined as $\mathcal{L} = \lambda_0 \mathcal{L}_2 + \lambda_1 \mathcal{L}_{\text{lpips}} + \lambda_2 \mathcal{L}_{\text{sim}}$, where $\mathcal{L}_2$ is the pixel-wise loss, $\mathcal{L}_{\text{lpips}}$ is the perceptual loss (Tov et al., 2021), $\mathcal{L}_{\text{sim}}$ is the loss measuring the cosine similarity, $\lambda_{0,1,2}$ are the coefficients weighing on the losses. Fig. 2 (right) shows that the approximate IID representation $\mathsf{Z}^i$ induced by $\phi$ (Restyle encoder with IN) achieves robust image reconstruction for RxRx1 and SCRC. See App. B, C for more hyper-parameter and result discussions.

**The Learned IID Representation in ERM**. After freezing the learned Restyle encoder $\phi$ described above, we integrate the $\phi$ induced IID representation $\mathsf{Z}^i$ to two standard (ResNet (He et al., 2016), DenseNet (Huang et al., 2017)) and two light-weight (MobileNet (Sandler et al., 2018), Mnas-Net (Tan et al., 2019)) backbones (See Fig. 2 (left)) that are widely used under the ERM framework. Due to the dimensional compatibility between $\mathsf{Z}^i_{m,0}, \ldots, \mathsf{Z}^i_{m,4}, \mathsf{Z}^i_{s,0}, \ldots, \mathsf{Z}^i_{s,13}$ and layer outputs of the compared backbones, this is implemented via adding the scaled 2-dim output ($\tilde{z}^i_{m,j} = \lambda_{m,j} z^i_{m,j}$ for $j = 0, \ldots, 4$) of morph blocks to the block of backbones, and via processing the 1-dim outputs ($\tilde{z}^i_s = \mathsf{Conv1d}(\mathsf{Cat}(z^i_{s,0}, \ldots, z^i_{s,13}))$) of stain blocks for latent vector concatenation (See also Fig. 2 (left bottom)), where $\lambda_{m,j}$ is a learnable scalar coefficient. Accordingly, the objective is to predict the class of genetic perturbation (RxRx1) and imCMS (SCRC) and defined as

| Optimal | Validation | ID Test | OOD Test |
|---|---|---|---|
| IRM | 7.31 (0.56) | 13.91 (1.29) | 11.90 (1.12) |
| CORAL | 19.47 (0.14) | 37.41 (0.36) | 30.91 (0.37) |
| GroupDRO | 15.17 (0.14) | 29.89 (0.12) | 24.35 (0.14) |
| ERM | 23.43 (0.36) | 47.20 (0.84) | 38.47 (0.55) |
| **Prop** | **24.14 (0.12)** | **48.95 (0.24)** | **39.31 (0.08)** |

| | | Validation | ID Test | OOD Test |
|---|---|---|---|---|
| ResNet50 | ERM | 23.43 (0.36) | 47.20 (0.84) | 38.47 (0.55) |
| | **Prop** | **24.14 (0.12)** | **48.95 (0.24)** | **39.31 (0.08)** |
| DenseNet121 | ERM | 23.45 (0.31) | 47.17 (0.45) | 37.86 (0.43) |
| | **Prop** | **23.58 (0.24)** | **48.52 (0.23)** | **38.30 (0.40)** |
| MobileNetV2 | ERM | **21.20 (0.36)** | 44.27 (0.79) | 34.65 (0.27) |
| | **Prop** | 21.15 (0.33) | **45.62 (0.33)** | **34.83 (0.37)** |
| MnasNet1_0 | ERM | 17.73 (0.36) | 35.82 (0.75) | 28.14 (0.69) |
| | **Prop** | **18.25 (0.28)** | **38.01 (0.61)** | **29.38 (0.51)** |

| | HEPG2 | | | HUVEC | | | RPE | | | U2OS | | |
|---|---|---|---|---|---|---|---|---|---|---|---|---|
| | Validation | ID Test | OOD Test | Valiation | ID Test | OOD Test | Validation | ID Test | OOD Test | Validation | ID Test | OOD Test |
| ERM | 22.09 (2.41) | 31.28 (3.79) | 25.82 (2.71) | **41.71 (5.20)** | 58.67 (5.89) | **45.76 (5.37)** | 19.51 (2.14) | 30.75 (4.07) | 25.88 (3.83) | **2.51 (0.35)** | 22.26 (2.89) | 11.12 (1.44) |
| Prop | **22.89 (2.18)** | **32.36 (3.55)** | **26.22 (2.61)** | 41.16 (5.15) | **59.97 (5.33)** | 45.73 (5.02) | **19.88 (2.04)** | **31.95 (3.81)** | **26.14 (3.58)** | 2.04 (0.32) | **23.65 (2.81)** | **11.52 (1.39)** |

Table 1: The main results of RxRx1. Top: The average classification accuracies with SD for optimally tuned (Optimal) compared methods (Left) and for ERM and proposed method (Prop) under the same backbones (Right). Bottom: The overall stratified accuracies with SD for ERM and Prop on 4 cell types: HEPG2, HUVEC, RPE, U2OS.

| | | SCRC0 | | SCRC1 | | SCRC2 | |
|---|---|---|---|---|---|---|---|
| | | Validation | OOD Test | Validation | OOD Test | Validation | OOD Test |
| | IRM | 65.13 (0.59) | 63.68 (1.22) | 70.18 (0.85) | 64.32 (0.57) | 67.26 (0.45) | 65.18 (0.13) |
| | CORAL | 62.75 (0.06) | 61.87 (0.99) | 68.36 (1.51) | 63.38 (0.88) | 65.92 (0.84) | 65.19 (0.32) |
| Optimal | GroupDRO | 63.64 (0.30) | 62.21 (0.91) | 69.36 (0.98) | 64.18 (0.29) | 67.58 (0.85) | 66.13 (0.81) |
| | ERM | 67.72 (0.80) | 66.34 (0.18) | 72.89 (0.37) | 66.89 (0.42) | 70.25 (1.04) | 66.59 (0.72) |
| | **Prop** | **67.87 (1.15)** | **66.81 (0.39)** | **73.44 (0.35)** | **66.91 (0.45)** | **70.64 (0.33)** | **66.97 (0.16)** |
| ResNet50 | ERM | **65.84 (0.87)** | 63.31 (1.00) | 72.78 (0.73) | 66.07 (0.83) | 67.40 (0.31) | 64.98 (0.75) |
| | **Prop** | 64.95 (0.67) | **64.35 (0.53)** | **73.07 (0.70)** | **66.23 (0.57)** | **68.62 (0.61)** | **65.88 (0.81)** |
| DenseNet121 | ERM | **64.31 (0.69)** | 63.74 (0.10) | 72.33 (0.87) | 65.64 (0.69) | 68.05 (0.80) | 65.54 (0.60) |
| | **Prop** | 64.17 (0.41) | **64.03 (0.21)** | **73.07 (0.40)** | **65.74 (0.74)** | **68.91 (0.44)** | **65.89 (0.88)** |
| MobileNetV2 | ERM | 67.72 (0.80) | 66.34 (0.18) | 72.89 (0.37) | 66.89 (0.42) | 70.25 (1.04) | 66.59 (0.72) |
| | **Prop** | **67.87 (1.15)** | **66.81 (0.39)** | **73.44 (0.35)** | **66.91 (0.45)** | **70.64 (0.33)** | **66.97 (0.16)** |
| MnasNet1_0 | ERM | 64.52 (0.99) | 63.61 (1.25) | **71.73 (0.57)** | **65.87 (0.71)** | 68.34 (0.47) | 65.50 (0.69) |
| | **Prop** | **65.12 (0.38)** | **64.56 (0.75)** | 71.25 (0.52) | 65.44 (1.44) | **68.77 (0.74)** | **65.94 (0.93)** |

| | | SCRC0 | | SCRC1 | | SCRC2 | |
|---|---|---|---|---|---|---|---|
| | | Validation | OOD Test | Validation | OOD Test | Validation | OOD Test |
| imCMS1 | ERM | 20.78 (4.57) | 20.76 (4.00) | 30.99 (7.21) | **32.71 (6.30)** | 23.44 (5.81) | **24.83 (7.34)** |
| | **Prop** | **23.12 (7.04)** | **22.60 (4.76)** | **33.85 (11.1)** | 31.52 (10.5) | **23.61 (5.56)** | 24.10 (6.85) |
| imCMS2 | ERM | **67.48 (4.05)** | 69.48 (4.01) | 69.46 (4.40) | 60.14 (3.69) | 52.40 (7.32) | 54.24 (6.44) |
| | **Prop** | 67.11 (3.78) | **69.62 (4.43)** | **70.14 (4.92)** | **60.81 (4.81)** | **53.32 (4.85)** | **55.52 (4.03)** |
| imCMS3 | ERM | 39.34 (8.00) | 39.03 (8.67) | 41.04 (8.88) | 33.25 (6.52) | **28.05 (4.09)** | 25.26 (4.12) |
| | **Prop** | **40.44 (6.52)** | **41.86 (6.84)** | **41.53 (8.33)** | **33.47 (6.69)** | 27.69 (5.44) | **25.39 (4.40)** |
| imCMS4 | ERM | **80.22 (3.77)** | 75.78 (3.64) | **82.84 (2.46)** | **79.57 (2.89)** | 87.49 (2.43) | 83.52 (2.95) |
| | **Prop** | 79.66 (3.26) | **76.05 (3.52)** | 82.71 (2.82) | 79.16 (3.10) | **88.42(1.64)** | **83.89 (2.04)** |

Table 2: The main results of SCRC. Left: The average classification accuracies with SD for optimally tuned (Optimal) compared methods and for ERM and proposed method (Prop) under the same backbones (Right). Right: The overall stratified accuracies with SD for ERM and Prop on imCMS1, 2, 3, 4 (Nguyen et al., 2021).

$\mathcal{L} = \lambda \mathcal{L}_{\text{crs}} + (1 - \lambda)\mathcal{L}_{\text{arc}}$, where $\mathcal{L}_{\text{crs}}$ is the cross-entropy loss, $\mathcal{L}_{\text{arc}}$ is the ArcFace loss (Deng et al., 2019), $\lambda$ is the coefficient balancing the losses. See App. E for more hyper-parameter discussions on SOTA baselines supported in WILDS and proposed method.

**Molecular Prediction Result**. Surprisingly, the ERM method outperforms the SOTA IRM (ERM + gradient), CORAL (ERM + covariance) and GroupDRO (ERM + worst-case group) in the experiments (See Tab. 1 and 2). More importantly, our proposed method (Prop: ERM + IID representation) achieves top classification accuracies compared to these optimally tuned baselines supported in WILDS for both ID (RxRx1) and OOD test data (SCRC, RxRx1). The consistent improvements under various backbones (Tab. 1 (right) and Tab. 2 (left)) confirm the reusability of learned IID representation. With further stratifying the results by cell types (Tab. 1 (bottom)) and imCMS classes (Tab. 2 (right)) we conclude that the proposed IID representation learning achieves superior results on OOD generalization tasks for RxRx1 and SCRC. Please see App. D for more ablation studies.

## 5. Conclusion

In this paper, we propose the IID representation learning and discuss its essential connection to causality. Experimental results on two biomedical datasets show that reusing learned IID representation can improve downstream molecular predictions in terms of OOD generalization. In future work, follow-up investigations from theoretical and biological viewpoints need be conducted to better understand the theoretical guarantee and underlying biological drivers of the IID representation.

## Acknowledgments

We would like to thank the Colorectal Cancer Research Group and gratefully acknowledge all members of the Translational Research Unit at the Institute of Pathology, University of Bern for excellent collaboration and provision of the CRC image dataset. We gratefully acknowledge the S:CORT consortium, a Medical Research Council stratified medicine consortium led by Prof. Tim Maughan at the University of Oxford, jointly funded by the MRC and CRUK; the current implementation of imCMS is a joint development of the S:CORT consortium at the University of Oxford in particular Prof. Jens Rittscher and Dr. Korsuk Sirinukunwattana at the Department of Engineering Science, Prof. Tim Maughan at the CRUK/MRC Oxford Institute for Radiation Oncology, and Dr. Enric Domingo at the Department of Oncology, University of Oxford with the Computational and Translational Pathology Group at the University of Zurich (Dr. Maxime Lafarge, Prof. Viktor Koelzer). The authors thank Anja Frei for data processing, Sonali Andani and Dr. Marta Nowak for insightful discussion. We gratefully acknowledge funding by the Promedica Foundation F-87701-41-01.

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

## Appendix A. Toy Experiments for Causal Variable Identification

Complementary to Sec. 4, we conduct toy experiments on the causal variable identification task (Sec. 3.1) for validating the proposed IID representation learning. This is done by following the experimental design of AICP (Gamella and Heinze-Deml, 2020) (Please see also `https://github.com/juangamella/aicp`). Specifically, we start the data simulation by creating a directed acyclic graph (DAG) endowed with vertices, edges and Gaussian noises, where the vertices of the DAG correspond to the variables $x_0, x_1, \ldots, x_l$ in Sec. 3.1. These specifications form a linear Gaussian SCM.

W.l.o.g. consider $x_0$ be the outcome variable and $X_{PA_0}$ be the set of $x_0$'s parent variables, under the assumption of without intervening on the outcome $x_0$, we implement do-interventions $e_{j=1,\ldots,l}$ independently via breaking the edges pointing to $x_j$ and letting $x_j := c$, which simulates the RCT setting described in Sec. 3.1. As specified in `https://github.com/juangamella/aicp`, we then collect $l$ batches of data samples that are randomly drawn from the SCM intervened with $e_{j=1,\ldots,l}$ resp. In the same manner as AICP (Gamella and Heinze-Deml, 2020), given such a dataset, our goal is to identify the set of parent variables of the outcome $x_0$.

Instead of a sophisticated auto-encoder proposed in Sec. 4, here we utilize a simple neural network $\phi' \circ \phi$, where $\phi'$ is a standard MLP layer, $\phi(x) = w \odot x$ is the element-wise multiplication of the input $x$ and binary penalty weights $w$ (initialized with 1). We propose to learn the projection map $\phi$ inducing identical interventional distribution among $(X_{PA_0}^{e_j}, x_0)$ for $j = 1, \ldots, l$, where $\phi$ should project $\{x_1, \ldots, x_l\}$ to $X_{PA_0}$. Noting that $\phi$ also induces independence among $(X_{PA_0}^{e_j}, x_0)$ for $j = 1, \ldots, l$ due to the independently intervened SCMs. Concretely, we train $\phi' \circ \phi$ for $l$ epochs with $\|.\|_2$ norm and iteratively penalize if $x_j \in X_{PA_0}$ holds true ($w_j$ of $\phi$ remains to be 1) for $j = 1, \ldots, l$ per epoch. Such penalty is conditioned on $\max_{j=1,\ldots,l} \text{FID}(\mu_j, \mu_{j^c})$, where FID is the Fréchet inception distance (Heusel et al., 2017), $\mu_j, \mu_{j^c}$ are the interventional distributions of $\|\phi' \circ \phi(x_1, \ldots, x_l) - x_0\|_2$ w.r.t. the data sampled from $\{e_j\}$ and $\{e_1, \ldots, e_l\} \setminus \{e_j\}$ intervened SCM(s) resp.

| | Jaccard Similarity (FWER) | | |
|---|---|---|---|
| | 2 Confounders | 1 Confounder | 0 Confounder |
| ICP | 0.318 (1.00) | 0.401 (0.84) | 1.00 (0.00) |
| NICP | 0.317 (1.00) | 0.406 (0.82) | 1.00 (0.00) |
| AICP | 0.438 (0.05) | 0.485 (0.11) | 1.00 (0.00) |
| **Proposed** | **0.909 (0.16)** | **0.926 (0.12)** | **1.00 (0.00)** |

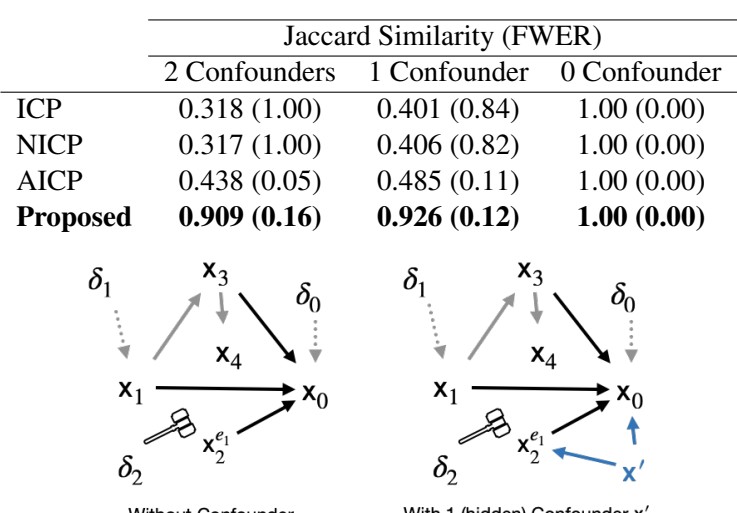

Table 3: Top: The results of causal variable identification for the toy experiments between ICPs and the proposed IID representation learning. Here, we report Jaccard Similarity (JS) and Family-wise error rate (FWER) (Gamella and Heinze-Deml, 2020) for quantitative comparison. Bottom: The visual illustration of SCMs with and without hidden confounder.

Then we compare the proposed method with ICP (Peters et al., 2016), NICP (Heinze-Deml et al., 2018) and AICP, all of which are developed upon the idea of identifying $X_{PA_0}$ via the intersection of sets of plausible causal variables. To better examine the robustness of compared methods, not only do we randomly choose 50 DAGs to re-run the experiments but additionally introduce 1 and 2 hidden confounder(s) for each DAG resp. As shown in Tab. 3, our proposed IID representation learning outperforms the ICPs especially with the inclusion of hidden confounder(s), in terms of better Jaccard Similarity (JS) and Family-wise error rate (FWER) (Gamella and Heinze-Deml, 2020) averaged on 50 DAGs,

$$JS(Z, X_{PA_0}) = \frac{|Z \cap X_{PA_0}|}{|Z \cup X_{PA_0}|}, \ FWER = \mathbb{P}(Z \not\subseteq X_{PA_0}), \text{where } Z = \phi(x_1, \dots, x_l) = (w_1 x_1, \dots, w_l x_l). \quad (3)$$

## Appendix B. Unsupervised Training of StyleGAN Decoder

Since there are not pre-trained StyleGAN (Karras et al., 2020) decoders available for the IID representation learning on RxRx1 and SCRC, we start the experiments with training StyleGAN in an unsupervised manner. Concretely, we take the widely-used PyTorch implementation `https://github.com/rosinality/stylegan2-pytorch` for training StyleGAN. Following the suggestions from WILDS, we only utilize the training data of RxRx1 and SCRC0,1,2 to learn four different StyleGAN models that can synthesize visually plausible microscopy images, while the validation and test data are held back during training. Due to the nature of moderate amount of training data, we follow the default configurations of StyleGAN training suggested in the repository except that we customize the training iterations to be 100k for all experiments, batch size to be 32 for RxRx1 and 16 for SCRC. Then we take advantage of the Distributed Data-Parallel (DDP) mechanism provided in PyTorch and train the StyGAN models on 4 A-100 GPUs and 2 A-100 GPUs for RxRx1 and SCRC respectively. We report the average Fréchet inception distance (FID) (Heusel et al., 2017) scores with SD obtained with four different random seeds for all the experiments in Tab. 4 and demonstrate the non-cherry-picked synthesized images in Fig. 3, 4, 5, 6. Noting that the large FID score for RxRx1 is resulted from comparing the total statistical difference on the ensemble of fluorescent medical images with more than 1000 classes of genetic perturbation, which differs from the common FID score computation in terms of a single class natural image generation (Karras et al., 2019, 2020).

| | FID |
|---|---|
| **RxRx1** | 21.27 (0.08) |
| **SCRC0** | 8.92 (0.14) |
| **SCRC1** | 8.59 (0.05) |
| **SCRC2** | 7.65 (0.16) |

Table 4: The average FID scores with SD achieved by StyleGAN on RxRx1 and SCRC0, 1, 2 obtained with four random seeds.

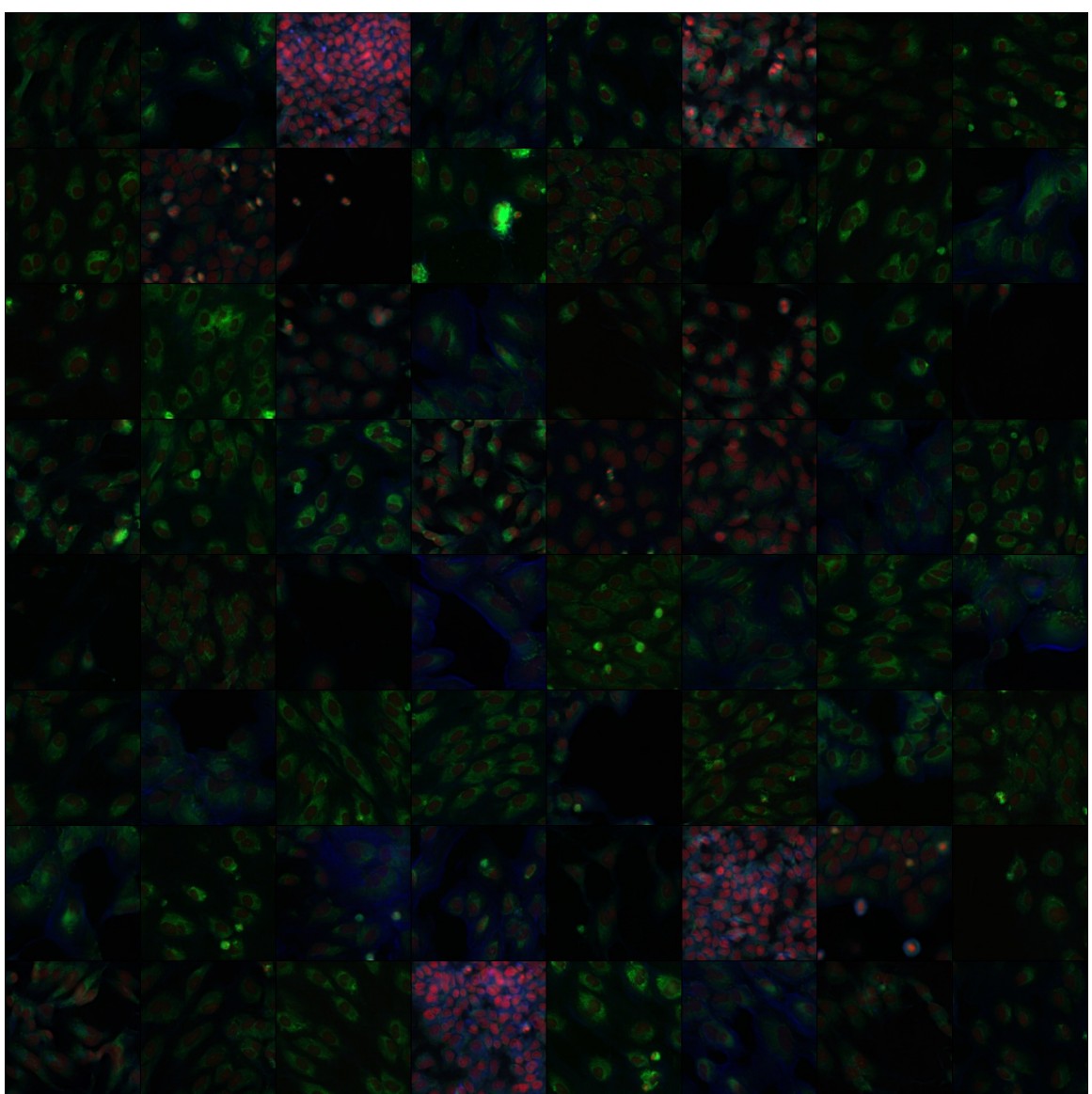

Figure 3: Nonexistent fluorescence images synthesized by StyleGAN learned with RxRx1 training data.

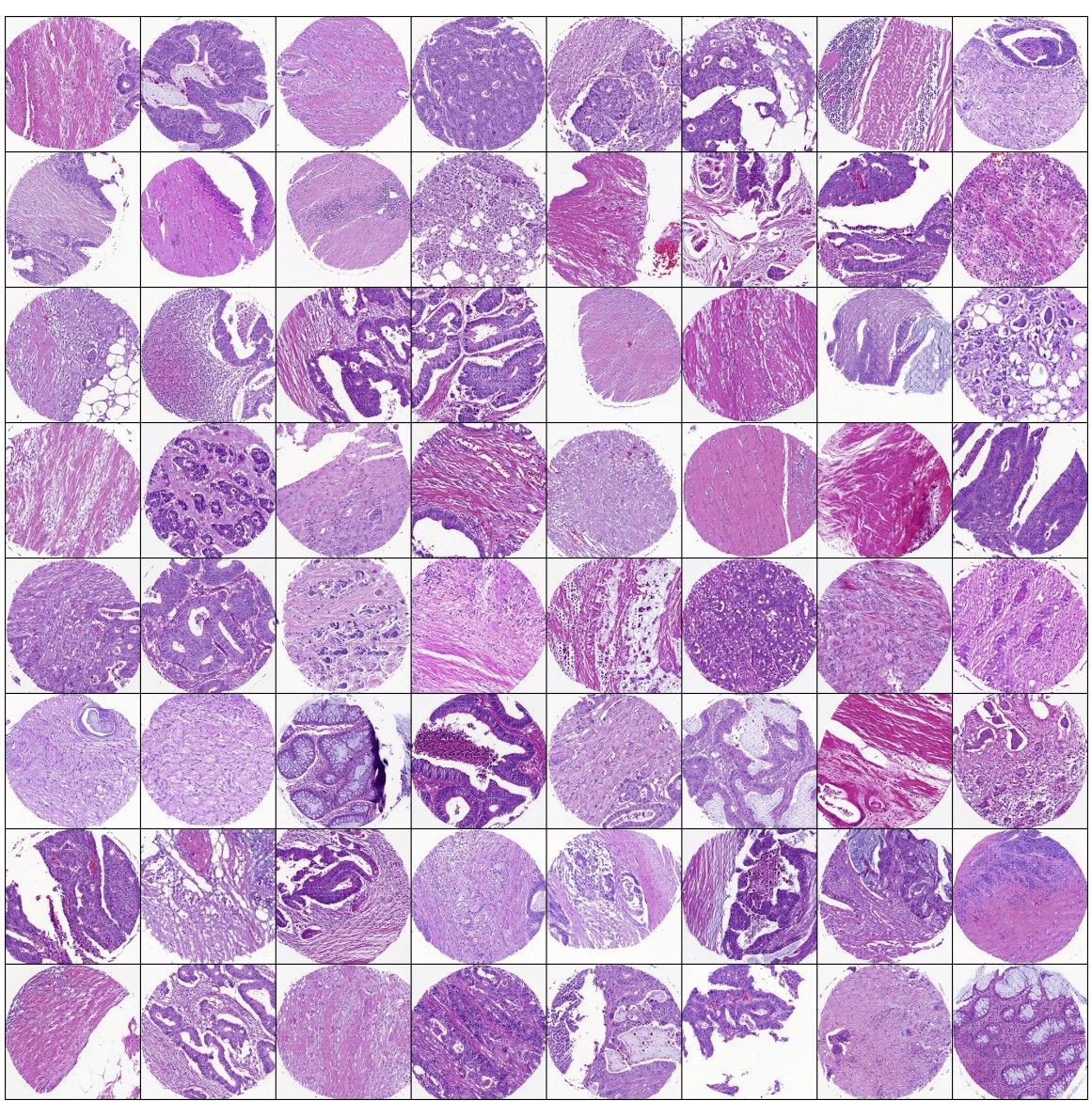

Figure 4: Nonexistent TMA images synthesized by StyleGAN learned with SCRC0 training data.

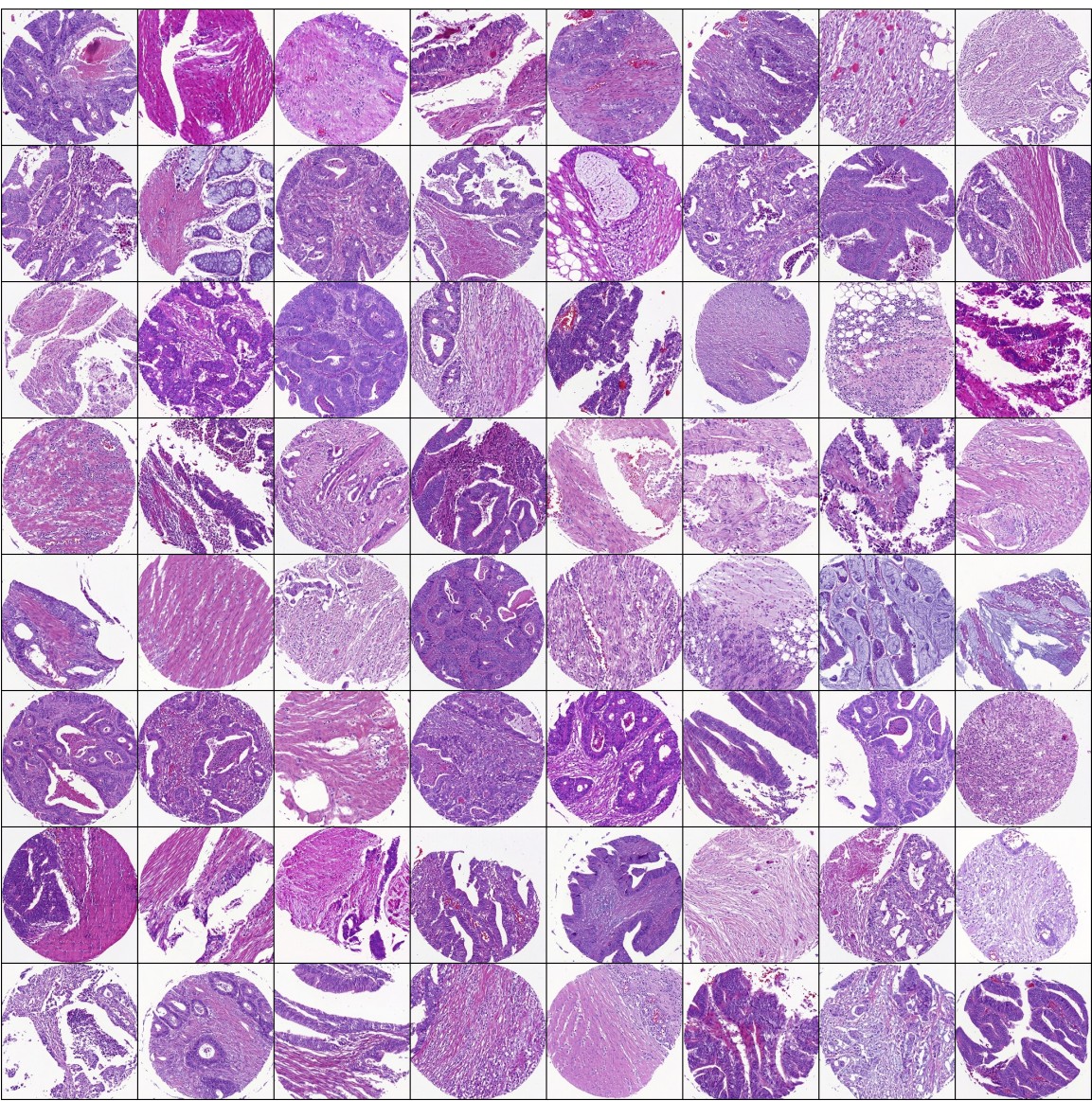

Figure 5: Nonexistent TMA images synthesized by StyleGAN learned with SCRC1 training data.

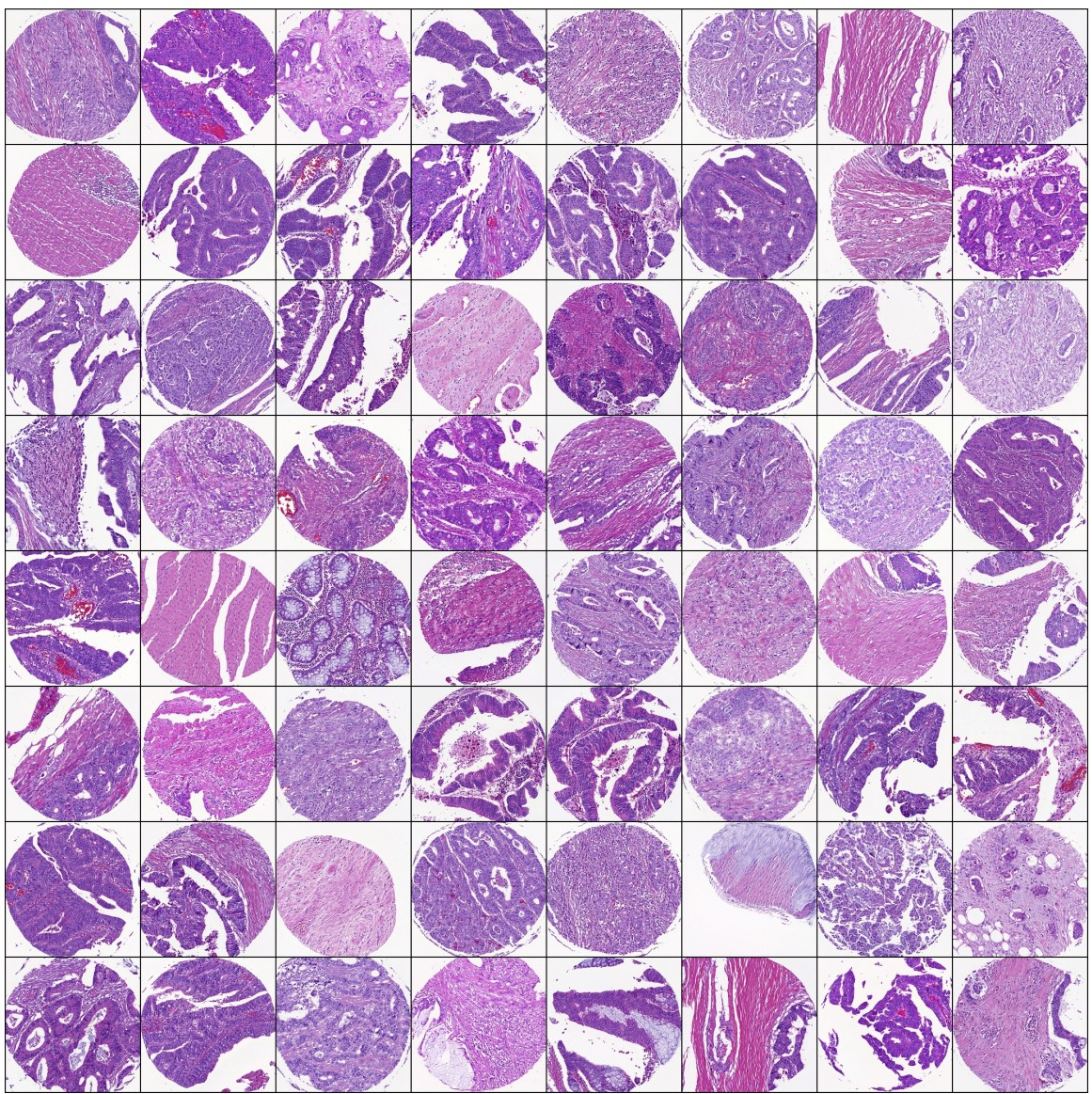

Figure 6: Nonexistent TMA images synthesized by StyleGAN learned with SCRC2 training data.

## Appendix C. Learning the Approximate IID Representation

To achieve faithful microscopy image reconstruction, we utilize the pre-trained StyleGAN decoder discussed in App. B and Restyle encoder (Alaluf et al., 2021) for learning the approximate IID representation. For the perceptual $\mathcal{L}_{\text{lpips}}$ (Zhang et al., 2018; Tov et al., 2021) and cosine similarity $\mathcal{L}_{\text{sim}}$ (Chen et al., 2020) loss of the reconstruction objective, we follow the default configuration introduced in Restyle encoder (Alaluf et al., 2021), *i.e.*, the $\mathcal{L}_{\text{lpips}}$ and $\mathcal{L}_{\text{sim}}$ are computed based on features extracted from the linear layer of the pre-trained AlexNet (Krizhevsky et al., 2012) and the MoCoV2 (Chen et al., 2020) pretrained ResNet50 (He et al., 2016) respectively, see also `https://github.com/yuval-alaluf/restyle-encoder` for more implementation details. Besides, by tuning on the validation data, it suffices to execute one step for iterative refinement and train all the experiments with 90k iterations. Lastly, the hyper-parameters $\lambda_{0,1,2}$ in the reconstruction objective are determined to be $1.5, 0.5, 0.5$ and $5, 0.2, 0.2$ for SCRC0,1,2 and RxRx1 respectively.

With computing the batch-wise statistics, the batch normalization (Ioffe and Szegedy, 2015) (BN) introduces unnecessary batch dependence between training data. Because of the element-wise affine operation applied on each image by default, the requirement that learning a function inducing identical distributions cannot be guaranteed by layer normalization (Ba et al., 2016) (LN). In combination of these observations and independent, approximately identically distributed $Z^i$ (Sec. 3.3) obtained via instance normalization (Ulyanov et al., 2016) (IN), we impose IN on the Restyle encoder (including the ResNet backbone). Under the same Restyle architecture, we run experiments and compare the reconstruction performance achieved between IN, BN (utilized in the default Restyle encoder), LN as well as group normalization (Wu and He, 2018) (GN). As a result, we experimentally justified the superiority of IN in terms of robust PSNR scores (See Tab. 5) and better visual qualities (See Fig. 7, 8, 9, 10).

| SCRC0 | Training | Validation | Test |
|---|---|---|---|
| BatchNorm (BN) | 16.30 (1.38) | 16.02 (1.32) | 15.93 (1.24) |
| LayerNorm (LN) | 16.75 (1.36) | 16.60 (1.29) | 16.52 (1.23) |
| GroupNorm (GN) | 17.13 (1.27) | 16.79 (1.23) | 16.68 (1.15) |
| **InstanceNorm (IN)** | **21.06 (1.35)** | **20.99 (1.22)** | **20.87 (1.17)** |

| SCRC1 | Training | Validation | Test |
|---|---|---|---|
| BatchNorm (BN) | 16.33 (1.39) | 16.21 (1.44) | 16.29 (1.41) |
| LayerNorm (LN) | 17.39 (1.25) | 17.28 (1.33) | 17.33 (1.30) |
| GroupNorm (GN) | 19.55 (1.11) | 19.47 (1.17) | 19.48 (1.16) |
| **InstanceNorm (IN)** | **21.44 (1.25)** | **21.35 (1.32)** | **21.36 (1.30)** |

| SCRC2 | Training | Validation | Test |
|---|---|---|---|
| BatchNorm (BN) | 18.37 (1.31) | 18.52 (1.38) | 18.49 (1.31) |
| LayerNorm (LN) | 18.86 (1.28) | 18.91 (1.37) | 18.89 (1.31) |
| GroupNorm (GN) | 17.82 (0.80) | 17.89 (0.82) | 17.89 (0.79) |
| **InstanceNorm (IN)** | **20.39 (1.20)** | **20.64 (1.37)** | **20.56 (1.29)** |

| RxRx1 | Training | Validation | Test |
|---|---|---|---|
| BatchNorm (BN) | 32.87 (4.35) | 29.80 (4.25) | 33.21 (4.19) |
| LayerNorm (LN) | 39.47 (3.06) | 37.76 (3.99) | 39.42 (2.55) |
| GroupNorm (GN) | 39.83 (3.49) | 37.49 (5.73) | 39.88 (2.96) |
| **InstanceNorm (IN)** | **43.46 (3.70)** | **41.68 (4.81)** | **43.13 (3.12)** |

Table 5: The average PSNR with SD achieved by four compared normalization methods under the same architecture of Restyle encoder and StyleGAN decoder.

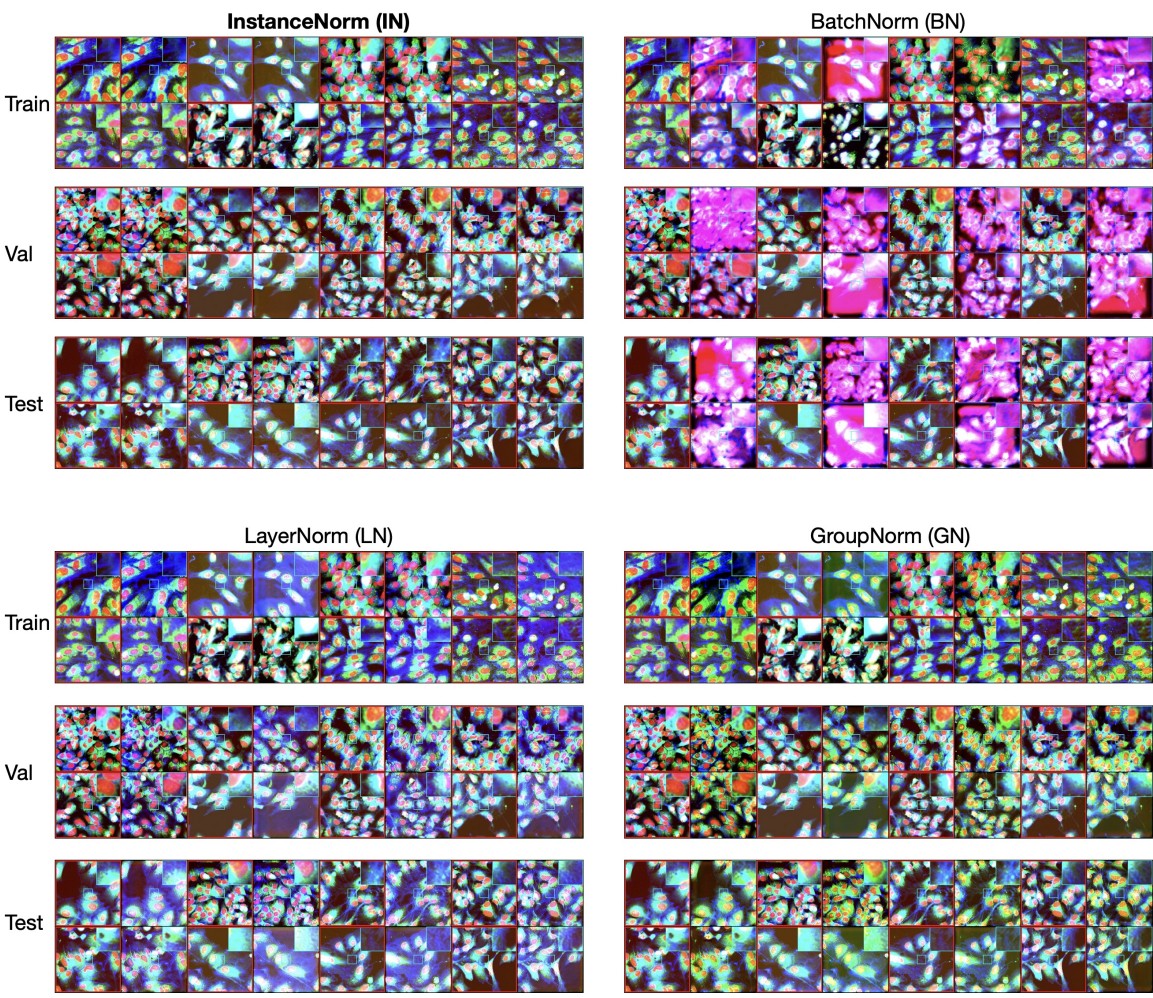

Figure 7: The RxRx1 visual comparison between ground-truth (red bounding box) and reconstructed images for Batch (BN), Layer (LN), Group (GN) and Instance (IN) normalization. Here, we normalize the ground-truth and reconstructed images along each channel for a clearer comparison. Please zoom in on the image details for better visualization.

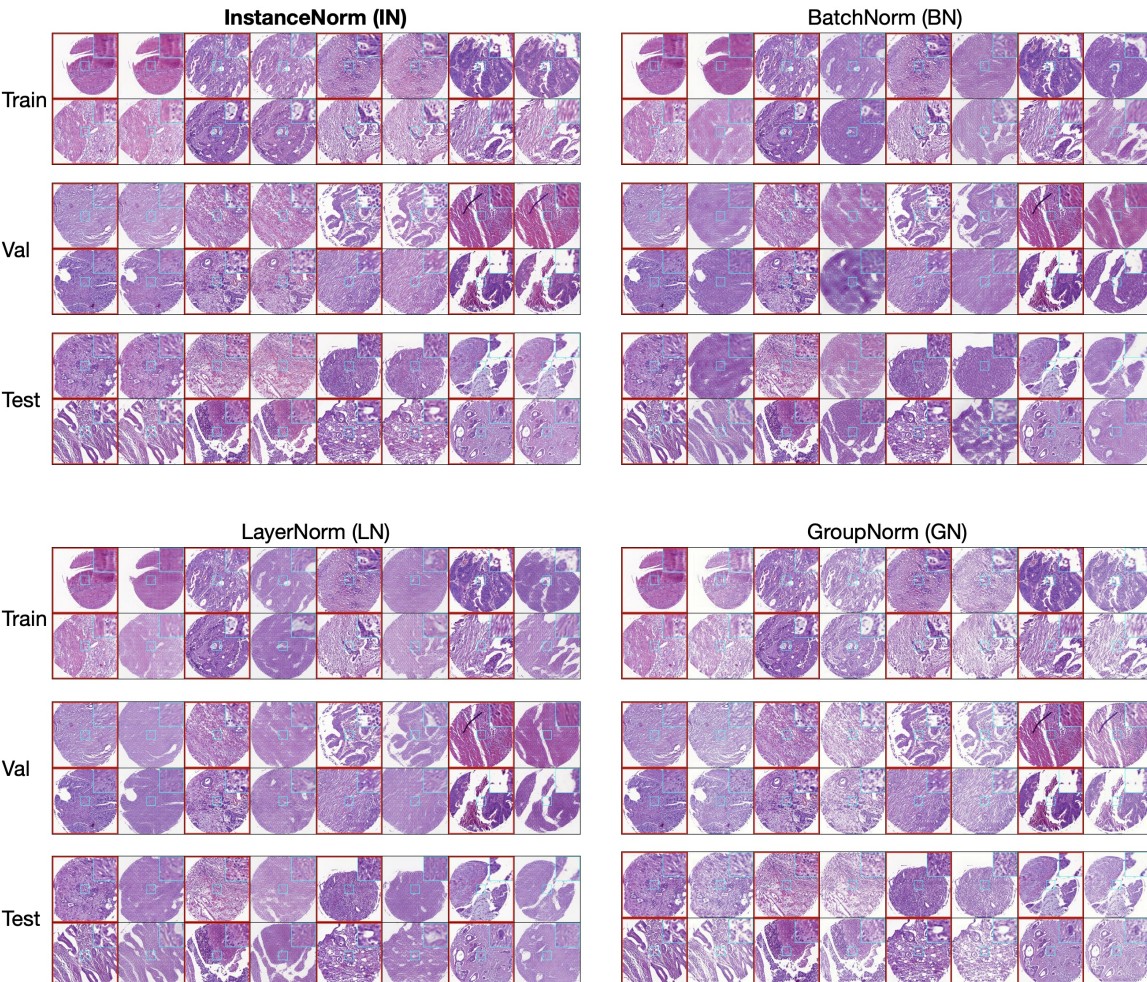

Figure 8: The SCRC0 visual comparison between ground-truth (red bounding box) and reconstructed images for Batch (BN), Layer (LN), Group (GN) and Instance (IN) normalization. Please zoom in on the image details for better visualization.

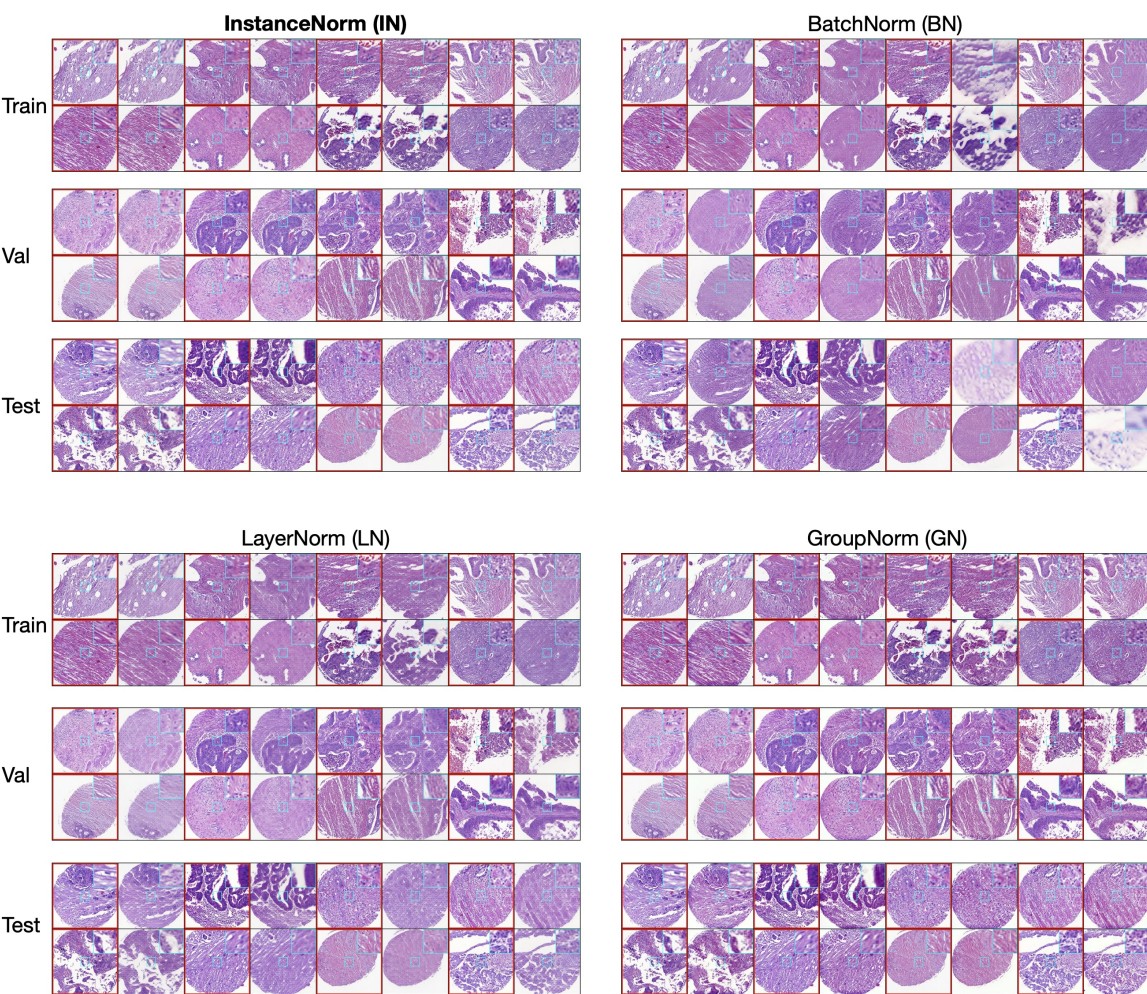

Figure 9: The SCRC1 visual comparison between ground-truth (red bounding box) and reconstructed images for Batch (BN), Layer (LN), Group (GN) and Instance (IN) normalization. Please zoom in on the image details for better visualization.

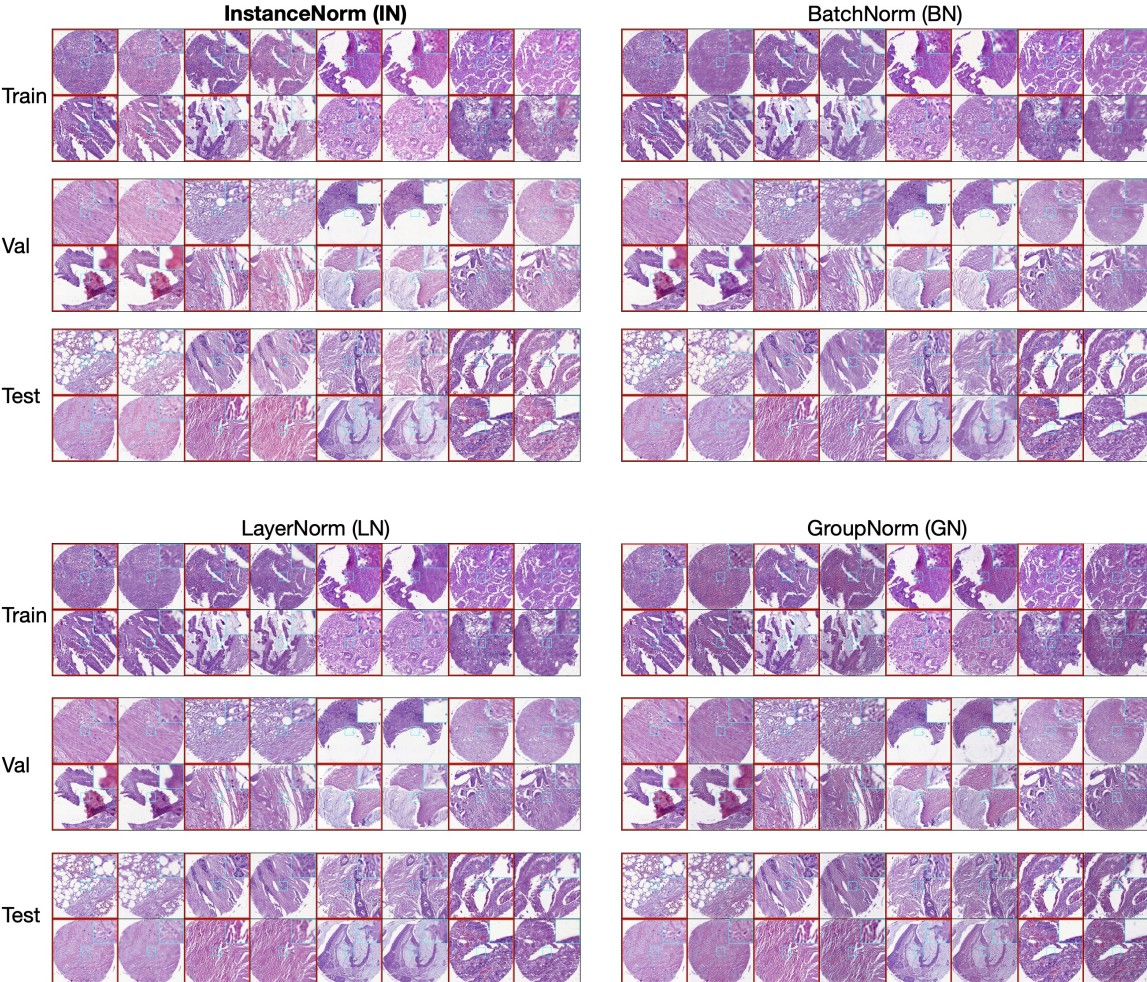

Figure 10: The SCRC2 visual comparison between ground-truth (red bounding box) and reconstructed images for Batch (BN), Layer (LN), Group (GN) and Instance (IN) normalization. Please zoom in on the image details for better visualization.

## Appendix D.  Ablation Studies on Stain and Morph blocks

When examining stain and morph blocks individually (See Fig. 11, 12), the takeaways are mixed. For RxRx1, the stand-alone stain blocks clearly contribute to the prediction improvement. This may be explained by the preanalytical variation in forms of batch-wise staining shift embedded in validation and test images. For SCRC, neither stain nor morph blocks bring clear quantitative improvements individually. Only by utilizing both of them can we robustify the OOD generalization.

| | | Validation | OOD Test | ID Test |
|---|---|---|---|---|
| RxRx1 | Stain | 23.77 (0.53) | 38.70 (0.69) | 49.01 (1.20) |
| | Morph | 23.39 (0.35) | 38.30 (0.54) | 47.39 (0.61) |
| SCRC0 | Stain | 64.67 (1.04) | 63.19 (0.41) | N/A |
| | Morph | 64.56 (0.51) | 64.05 (0.61) | N/A |
| SCRC1 | Stain | 73.07 (0.57) | 65.67 (0.56) | N/A |
| | Morph | 72.85 (0.15) | 65.58 (1.15) | N/A |
| SCRC2 | Stain | 68.08 (0.43) | 65.51 (0.53) | N/A |
| | Morph | 68.34 (0.80) | 64.68 (0.79) | N/A |

Figure 11: The ablation studies of utilizing stain or morph blocks individually.

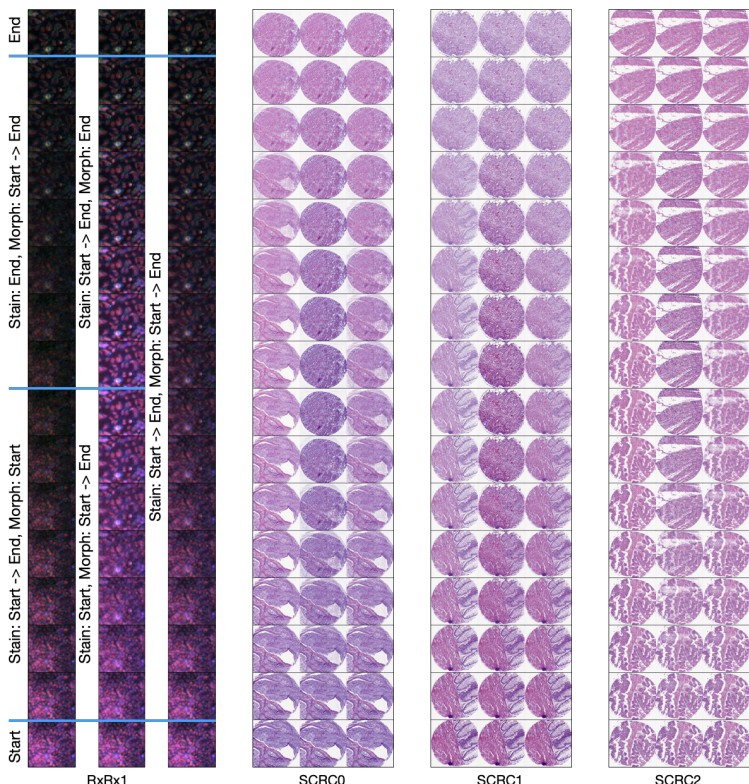

Figure 12: The RxRx1 and SCRC0,1,2 visualization of interpolating the outputs of stain and morph blocks simultaneously, interpolating stain outputs while freezing morph ones and vice-versa. Please zoom in on the image details for better visualization.

## Appendix E. The learned IID Representation in ERM

To enable reproducibility and for comparison to the SOTA methods, we utilize the WILDS repository `https://github.com/p-lambda/wilds.git` to run the experiments. Precisely, we call the data loader functions of RxRx1 implemented in WILDS and write the corresponding data loader functions for SCRC following the WILDS coding style. Except that we introduce the CutMix (Yun et al., 2019) as a complement to the standard augmentation methods supported in WILDS, we do not use additional techniques such as fusing the outputs from several rotated inputs or from multiple models to boost the performance for compared methods. During the training, we do not feed the validation and test data to the model, the validation data is only used for hyper-parameter tuning. Accordingly, all compared methods are well tuned on the hyper-parameters with the careful selection of augmentations, backbones, *etc*.

For ERM, we determine the optimal $\lambda$ to be 0.8 and $\lambda_{\mathsf{CutMix}}$ to be 1 for RxRx1 and SCRC, ResNet50/DenseNet121 for RxRx1 and MobileNetV2 for SCRC. For IRM and CORAL, we determine the optimal $\lambda$ to be 1, the backbone to be MobileNetV2 and $\lambda_{\mathsf{CutMix}} = 0$ for both RxRx1 and SCRC. In terms of GroupDRO, the configurations are the same to IRM and CORAL except that with utilizing DenseNet121 it achieves competitive results to MobileNetV2 in SCRC experiments. As to the proposed method (Prop), the optimal results are obtained by $\lambda = 0.8, \lambda_{\mathsf{CutMix}} = 1$ for both RxRx1 and SCRC, as well as ResNet50 for RxRx1 and MobileNetV2 for SCRC. Complementary to the Tab. 1 and 2 in the main manuscript, we present detailed results for all compared methods with respect to the same backbones in Tab. 6 and 7.

| ResNet50 | Validation | ID Test | OOD Test |
|---|---|---|---|
| IRM | 5.60 (0.40) | 8.20 (1.10) | 9.90 (1.40) |
| CORAL | 18.50 (0.40) | 28.40 (0.30) | 34.00 (0.30) |
| GroupDRO | 15.20 (0.10) | 28.10 (0.30) | 23.00 (0.30) |
| ERM | 23.43 (0.36) | 47.20 (0.84) | 38.47 (0.55) |
| **Prop** | **24.14 (0.12)** | **48.95 (0.24)** | **39.31 (0.08)** |

| DenseNet121 | Validation | ID Test | OOD Test |
|---|---|---|---|
| IRM | 6.71 (0.63) | 12.87 (0.64) | 10.74 (0.80) |
| CORAL | 14.91 (0.18) | 28.24 (0.37) | 24.05 (0.31) |
| GroupDRO | 10.89 (0.33) | 21.25 (0.57) | 17.79 (0.46) |
| ERM | 23.45 (0.31) | 47.17 (0.45) | 37.86 (0.43) |
| **Prop** | **23.58 (0.24)** | **48.52 (0.23)** | **38.30 (0.40)** |

| MobileNet_V2 | Validation | ID Test | OOD Test |
|---|---|---|---|
| IRM | 7.31 (0.56) | 13.91 (1.29) | 11.90 (1.12) |
| CORAL | 19.47 (0.14) | 37.41 (0.36) | 30.91 (0.37) |
| GroupDRO | 15.17 (0.14) | 29.89 (0.12) | 24.35 (0.14) |
| ERM | **21.20 (0.36)** | 44.27 (0.79) | 34.65 (0.27) |
| **Prop** | 21.15 (0.33) | **45.62 (0.33)** | **34.83 (0.37)** |

| MnasNet1_0 | Validation | ID Test | OOD Test |
|---|---|---|---|
| IRM | 7.18 (0.24) | 13.32 (0.47) | 11.48 (1.02) |
| CORAL | 13.06 (0.73) | 25.39 (1.44) | 20.35 (0.77) |
| GroupDRO | 14.02 (0.41) | 27.52 (0.36) | 21.97 (0.36) |
| ERM | 17.73 (0.36) | 35.82 (0.75) | 28.14(0.69) |
| **Prop** | **18.25 (0.28)** | **38.01 (0.61)** | **29.38 (0.51)** |

Table 6: The average classification accuracies with SD of RxRx1 that are obtained with four different backbones for all compared methods.

| ResNet50 | SCRC0 | | SCRC1 | | SCRC2 | |
|---|---|---|---|---|---|---|
| | Validation | OOD Test | Validation | OOD Test | Validation | OOD Test |
| IRM | 61.22 (0.72) | 60.72 (0.50) | 69.40 (0.80) | 63.34 (0.59) | 64.48 (0.91) | 63.42 (0.48) |
| CORAL | 60.62 (0.44) | 60.42 (0.56) | 68.14 (0.32) | 64.17 (0.44) | 66.10 (0.48) | 63.86 (0.90) |
| GroupDRO | 61.04 (0.77) | 60.55 (0.25) | 69.88 (0.93) | 64.49 (0.55) | 66.10 (0.68) | 64.07 (0.50) |
| ERM | **65.84 (0.87)** | 63.31 (1.00) | 72.78 (0.73) | 66.07 (0.83) | 67.40 (0.31) | 64.98 (0.75) |
| **Prop** | 64.95 (0.67) | **64.35 (0.53)** | **73.07 (0.70)** | **66.23 (0.57)** | **68.62 (0.61)** | **65.88 (0.81)** |

| DenseNet121 | SCRC0 | | SCRC1 | | SCRC2 | |
|---|---|---|---|---|---|---|
| | Validation | OOD Test | Validation | OOD Test | Validation | OOD Test |
| IRM | 62.39 (1.02) | 62.87 (0.60) | 70.33 (0.86) | 64.31 (0.72) | 66.64 (0.51) | 64.72 (1.00) |
| CORAL | 61.79 (0.94) | 61.36 (0.53) | 69.47 (0.80) | 63.88 (0.17) | 66.21 (0.82) | 64.94 (0.60) |
| GroupDRO | 61.97 (0.88) | 61.55 (0.39) | 70.36 (1.32) | 65.02 (1.19) | 66.10 (0.48) | 65.75 (0.59) |
| ERM | **64.31 (0.69)** | 63.74 (0.10) | 72.33 (0.87) | 65.64 (0.69) | 68.05 (0.80) | 65.54 (0.60) |
| **Prop** | 64.17 (0.41) | **64.03 (0.21)** | **73.07 (0.40)** | **65.74 (0.74)** | **68.91 (0.44)** | **65.89 (0.88)** |

| MobileNetV2 | SCRC0 | | SCRC1 | | SCRC2 | |
|---|---|---|---|---|---|---|
| | Validation | OOD Test | Validation | OOD Test | Validation | OOD Test |
| IRM | 65.13 (0.59) | 63.68 (1.22) | 70.18 (0.85) | 64.32 (0.57) | 67.26 (0.45) | 65.18 (0.13) |
| CORAL | 62.75 (0.06) | 61.87 (0.99) | 68.36 (1.51) | 63.38 (0.88) | 65.92 (0.84) | 65.19 (0.32) |
| GroupDRO | 63.64 (0.30) | 62.21 (0.91) | 69.36 (0.98) | 64.18 (0.29) | 67.58 (0.85) | 66.13 (0.81) |
| ERM | 67.72 (0.80) | 66.34 (0.18) | 72.89 (0.37) | 66.89 (0.42) | 70.25 (1.04) | 66.59 (0.72) |
| **Prop** | **67.87 (1.15)** | **66.81 (0.39)** | **73.44 (0.35)** | **66.91 (0.45)** | **70.64 (0.33)** | **66.97 (0.16)** |

| MnasNet1_0 | SCRC0 | | SCRC1 | | SCRC2 | |
|---|---|---|---|---|---|---|
| | Validation | OOD Test | Validation | OOD Test | Validation | OOD Test |
| IRM | 44.03 (2.45) | 43.30 (1.98) | 58.31 (1.65) | 55.48 (1.38) | 59.19 (0.45) | 56.88 (0.54) |
| CORAL | 45.31 (5.32) | 43.71 (4.91) | 58.20 (1.77) | 55.38 (1.10) | 59.04 (0.28) | 56.54 (0.15) |
| GroupDRO | 46.48 (4.57) | 45.81 (3.41) | 57.86 (1.61) | 55.10 (1.55) | 58.93 (0.10) | 56.41 (0.08) |
| ERM | 64.52 (0.99) | 63.61 (1.25) | **71.73 (0.57)** | **65.87 (0.71)** | 68.34 (0.47) | 65.50 (0.69) |
| **Prop** | **65.12 (0.38)** | **64.56 (0.75)** | 71.25 (0.52) | 65.44 (1.44) | **68.77 (0.74)** | **65.94 (0.93)** |

Table 7: The average classification accuracies with SD of SCRC that are obtained with four different backbones for all compared methods.

