# OpenReview forum: "Towards IID representation learning and its application on biomedical data"
_MIDL.io/2022/Conference — MIDL 2022_

### Official Review · Reviewer_Wa34 · 2022-01-25

**Confidence:** 3
**Preliminary Rating:** 4
**Recommendation:** Poster

**Summary:**

This paper proposes a novel view of the IID effect in real world applications. It considers an intervention-induced IID as a new causal relationship to learn. This is different from existing causality learning and also different from previous work which simply takes the IID assumption. This I found very interesting. As a proof-of-concept, an empirical study on OOD prediction is presented in the paper.

**Strengths:**

- The presented idea seems novel, quite interesting and is justified in practice.
- A proof-of-concept empirical study is provided to validate the idea.
- Code is promised. This will help readers understand the method.

**Weaknesses:**

- The validation is directly on real world problems. I think it is quite necessary to use some synthetic data to validate the proposed method first. This has several benefit: (1) the method is more transparent without sophisticated neural networks involved; (2) a synthetic experiment can help better understand the main idea.

**Deanonymize Review:**

no

**Final Rating After The Rebuttal:**

4: Weak Accept

**Justification Of The Final Rating:**

Overall I think the motivation of the paper is quite clear and convincing. Different concerns raised by reviewers seemed to have been addressed in the rebuttal. Therefore I recommend the paper to be accepted.

**Paper Type:**

methodological development

**Questions To Address In The Rebuttal:**

I think the paper has great potential. But I think it would be necessary to provide synthetic experiments to better validate the proposed method, and also to provide better illustration of the main idea. I will be happy to increase my scores if this is provided during rebuttal.

**Special Issue:**

yes

---

### Official Review · Reviewer_2ZBQ · 2022-01-26

**Confidence:** 3
**Preliminary Rating:** 4
**Recommendation:** Poster

**Summary:**

The authors propose to model out-of-distribution (OOD) data as IID, which they termed IID representation learning. Here, the authors utilize an encoder-decoder network (based on standard architectures) to learn the IID representation and then combine this result with empirical risk minimization (ERM) for prediction. Experiments are provided using two datasets: RxRx1 to predict genetic perturbations from fluorescence microscopy and Swiss Colorectal Cancer (SCRC) to classify molecular subtypes of CRC from tissue microarray images. These prediction results are compared to SOTA baseline methods from WILDS, a benchmark of in-the-wild distribution shifts spanning diverse data.

**Strengths:**

• Experimental setup follows the WILDS benchmark guidelines to promote reproducibility and comparison to SOTA methods.

• Prediction performance comparison to four other baseline SOTA methods from WILDS is strong.

• Ablation studies to examine the effect of different encoder backbones and to test for the contributions of the stain and morphology block is rigorous.

• Background and prior work are rigorously covered and motivate the problem of out-of-distribution (OOD) generalization.

• This is a very well written paper.

• Code is publicly available on GitHub.


**Weaknesses:**

• The performance results are not substantially better than the benchmark ERM method in the two tested datasets.

• The paper does a good job introducing some complex definitions of causality and IID, but the graphical visualization (Fig. 1) is a little challenging to understand, especially how the hammer icon fits in because the concept of inability to implement an intervention was not really discussed in Sec. 3’s “Do-Intervention” subsection.


**Deanonymize Review:**

no

**Detailed Comments:**

Fig 1.: As mentioned above in the Weaknesses, I think the inclusion of the hammer icon is a little challenging to understand because the ideas of not being able to implement a particular intervention due to ethical reasons (which makes sense in randomized clinical trials) is not really discussed in Sec. 3’s “Do-Intervention” subsection. Reworking the figure a little or adding a little more descriptive text to this subsection could help clarify things for readers.

**Final Rating After The Rebuttal:**

4: Weak Accept

**Justification Of The Final Rating:**

Thank you for your revisions and the additional synthetic validation experiments. I feel that the presented manuscript would be of interest to the MIDL community and therefore maintain my original rating.

**Paper Type:**

both

**Questions To Address In The Rebuttal:**

Overall, I think the authors present a good manuscript that is backed by a solid set of experimental results with good reproducibility. This is an important topic in machine in general and has good relevance to the MIDL community.

**Special Issue:**

no

---

### Official Review · Reviewer_4zXw · 2022-01-27

**Confidence:** 4
**Preliminary Rating:** 2

**Summary:**


This work challenges the common IID assumption in datasets and proposes learning representations that induce independent & identical distribution (in the representations). They argue that the problem of learning causal mechanisms can be posed as learning "IID representations."

The authors applied ideas developed in this paper for generalization on out-of-distribution data. Their representation learning framework consists of an auto-encoder and uses reconstruction objectives. The encoder is frozen once the representations are learned, and a classifier is trained for downstream tasks. Using this setup, the authors demonstrate good generalization performance on two biomedical imaging datasets compared to ERM and other methods for OOD generalizations.


**Strengths:**


The paper tries to incorporate/identify causal factors in representation learning which is very relevant, especially for the medical field.

The proposed approach outperforms others and generalizes better.


**Weaknesses:**

**The paper is tough to follow and is vague/wrong in many places.**
- Abstract:
   - "task-relevant function that induces IID" -> inducing IID over what?
   - Also, the representation learning approach uses a reconstruction objective that is not task-dependent. What do authors mean by the term task-relevant?

- Introduction:
  - The term "IID" is freely used without specifying which variables are involved.
  - After reading the introduction, how causality will improve OOD generalization is unclear to me. I think it will help to elaborate on how causality and representation learning are related or what is the connection between the two.

- Sec 2 and 3:
   - The definitions and notations here are very hard to follow.
   - In Fig 1 and sec 3, authors call $\delta$ as the "confounder," but it is not a confounder (see [Counfounding](https://www.wikiwand.com/en/Confounding)). In particular, confounders influence both dependent and independent variables, whereas $\delta$ only influences the dependent variable.
    - What do grey edges indicate in Fig 3?

- Sec 4:
  - It is unclear how definitions in Sec 2 and 3 relate to experiments or inspire the authors' proposed framework.
  - The assumptions on the type of shift are not mentioned explicitly. For examples, which distributions (p(x), p(x|y), p(y|x) etc.) are expected to be same across domains. Please specify the assumptions. It is also unclear why OOD generalization is the best way to examine causal representations.
  - It is also unclear how the independence is induced and on what variables. Def 1 suggests inducing IID distribution over $z_i$ across samples, but it seems the autoencoder induces IID distribution between all latent factors.

**Results:**
   - Why does ERM achieve worse results on the in-distribution dataset? This is counterintuitive.
   - Above observation raises a second question: what data was used to train the encoder? Did the authors use the samples from the OOD test set to train the encoder?


**Deanonymize Review:**

no

**Final Rating After The Rebuttal:**

3: Borderline

**Justification Of The Final Rating:**

Thank you for revising the paper and for additional experiments and clarification. The updates have clarified much of my concerns. So I am raising my score by 1.

However, it is still unclear how IID constraints are enforced or the distribution over representations can be made IID using Instance Norm only. Using Instance Norm will ensure that the representation lies on the surface of a high-dimensional sphere, but that does not ensure IID distribution. Similarly, it appears that the authors have employed a different technique for toy experiments and not Instance Normalization. Due to this issue, I am keeping my final rating borderline.

**Paper Type:**

methodological development

**Questions To Address In The Rebuttal:**

See comments in the weakness section.

In particular, it would help:
- if the authors clarified how Sec 2 and 3 are related to experiments
- if the authors can improve the exposition of Sec 2 and 3 by running toy examples
- If assumptions regarding data distribution are made clear.

**Special Issue:**

no

---

### Meta-Review · Area_Chair_BEyU · 2022-02-21

**Recommendation:** Accept (Poster)
**Confidence:** 3

**Metareview:**

Strengths of this paper include 1) experimental setup and extensive validation and 2) novel and interesting approach to incorporate causality into representation learning. However, some problems with clarity and questions of change in approach when adding the new toy experiments reduced the ratings of this work.

---

### Decision · Program_Chairs · 2022-02-28

Accept